



# Cosmogenic nuclide dating of two stacked ice masses: Ong Valley, Antarctica

Marie Bergelin[1], Jaakko Putkonen[1], Greg Balco[2], Daniel Morgan[3], Lee B. Corbett[4] and Paul R. Bierman[4]

[1]Harold Hamm School of Geology and Geological Engineering, University of North Dakota, Grand Forks, ND, 58202
[2]Berkeley Geochronology Center, Berkeley, CA
[3]Department of Earth and Environmental Science, Vanderbilt University, Nashville,
[4]Rubenstein School of the Environment and Natural Resources, University of Vermont/National Science Foundation Community Cosmogenic Facility.

*Correspondence to*: Marie Bergelin (marie.bergelin@und.edu)

**Abstract.** We collected a debris rich ice core from a buried ice mass in Ong Valley, located in Transantarctic mountains in Antarctica. We measured cosmogenic nuclide concentrations in quartz obtained from the ice core to determine the age of the buried ice mass and infer the processes responsible for the emplacement of the debris currently overlaying the ice. Such ice masses are valuable archives of paleoclimate proxies; however, the preservation of ice beyond 800 kyrs is rare and therefore much effort has been recently focused on finding ice that is older than 1 Ma. In Ong Valley, the large, buried ice mass has been previously dated at > 1.1 Ma. Here we provide a forward model that predicts the accumulation of the cosmic-ray produced nuclides $^{10}$Be, $^{21}$Ne, and $^{26}$Al in quartz in the englacial and supraglacial debris and compare the model predictions to measured nuclide concentrations in order to further constrain the age. Large downcore variation in measured cosmogenic nuclide concentrations suggests that the englacial debris is sourced both from subglacially-derived material and recycled paleo surface debris that has experienced surface exposure prior to entrainment. We find that the upper section of the ice core is 2.95 +0.18/-0.22 Myrs. The average ice sublimation rate during this time period is 22.86 +0.10/-0.09 m Myr$^{-1}$, and the surface erosion rate of the debris is 0.206 +0.013/-0.017 m Myr$^{-1}$. Burial dating of the recycled paleo surface debris suggests that the lower section of the ice core belongs to a separate, older ice mass which we estimate to be 4.3-5.1 Myrs old. The ages of these two stacked, separate ice masses can be directly related to glacial advances of the Antarctic ice sheet and potentially coincide with two major global glaciations during the early and late Pliocene epoch when global temperatures and $CO_2$ were higher than present. These ancient ice masses represent new opportunities for gathering ancient climate information.



# 1 Introduction

Ice cores from glaciers and ice sheets are used as an archive for paleoclimate proxies, including: atmospheric gases, chemical
compounds, and airborne particles (Dansgaard et al., 1969; Fredskild and Wagner, 1974; Castellano et al., 2004; Willerslev
et al., 2007); however, the potential age of ice core records is limited by the fact that ice sheets are subject to deformation,
ice flow, and basal melting. The oldest ice that has been recovered from the thickest parts of the Antarctic ice sheets is
800,000 years old (Jouzel et al., 2007). Although it is hypothesized that ice up to 1-1.5 Ma may also exist at great depth in
the ice sheet (Fischer et al., 2013), recovering this ice would be a complex and costly endeavor. Therefore, we currently lack
archives of climate information that extend beyond ~0.8 Ma.

Bare ice is, in general, thermodynamically unstable under typical atmospheric pressure-temperature conditions and therefore
prone to melt and/or sublimate. However, there exist regions of topographically constrained, extremely slow ice flow in
which ice up to 2.7 Ma has been recovered near the surface (Yan et al., 2019). There are also several areas within the
Transantarctic Mountains where glacial ice is covered by supraglacial debris. A thick debris cover thermally insulates the ice
surface and provides a physically plausible means of preserving near-surface ice for long periods. For example, Sugden et al.
(1995) found glacier ice in Beacon Valley underlying a supraglacial debris containing an 8.1 Ma volcanic ash, leading them
to conclude that the ice is older than 8.1 Ma. However, the antiquity of the Beacon Valley ice has been questioned on the
basis of data suggesting that ice lost to sublimation is dynamically replaced by ice flow from upstream glaciers, resulting in a
situation where relatively young ice underlies relatively old debris (Van Der Wateren and Hindmarsh, 1995; Ng et al., 2005;
Hindmarsh et al., 1998; Stone et al., 2000). The lack of ice older than ~1 Ma severely limits our direct paleoclimate record
and creates uncertainties when modeling future climate predictions which include modeled configuration of the past
Antarctica Ice Sheet (Bulthuis et al., 2019; Noble et al., 2020). This is particularly important during the Pliocene epoch
(Dolan et al., 2018; Haywood et al., 2009), in which global surface temperatures and $CO_2$ levels were higher than present
(Pagani et al., 2010; Seki et al., 2010) and which is considered analog for current anthropogenic warming.

In Ong Valley, Miller Range, Transantarctic Mountains, a mass of glacier ice at least several tens of meters thick is found
buried underneath <1 m of supraglacial debris. Cosmogenic nuclide measurements from the supraglacial debris  suggest an
age of >1.1 Ma, but most likely >1.8 Ma (Bibby et al., 2016). We collected a 944 cm long ice core from this buried ice mass
and use concentrations of $^{10}$Be, $^{21}$Ne, and $^{26}$Al from the englacial debris to further constrain the age of the ice mass.

Our goal is to determine the age of the ice, understand its overall geologic history, and evaluate its potential use as a
paleoclimate archive. We present a novel dating application of cosmogenic nuclides which aims to quantify a complex





exposure history of this buried ice mass. By comparing measured cosmogenic nuclide concentrations from the englacial and
supraglacial debris with modeled concentrations, the nuclide inventory inherited from prior exposure can be distinguished
from that produced after ice emplacement. We then apply a cosmogenic-nuclide burial dating method to the inherited
inventory as an age constraint. We show that two sections of the ice core contain recycled surface debris that can be burial-
dated. The upper section is ~3 Ma, which we interpret as the emplacement age of the bulk of the buried ice. The lower
section has a significantly older burial age of > 4 Ma, and we interpret it as a portion of an older ice mass either *in situ* or
transported during emplacement of the younger ice.

## 2 Study area

Ong Valley is a ~1.5 km wide and ~7 km long glacial valley located in the Miller Range of the central Transantarctic
Mountains, Antarctica (83.25°S, 157.72°E). The current valley floor gradually rises from an elevation of 1500 m above sea
level (masl) to 1700 masl at the valley head. Over the span of one year (2011), the recorded air temperature in the valley
ranged between -49.0°C and -4.0°C, with mean of -23.9°C (Bibby et al., 2016). In the head of the valley is a small alpine
glacier, and the valley mouth is blocked by a 2 km wide exposed glacial ice front of the Argosy glacier (Fig. 1).

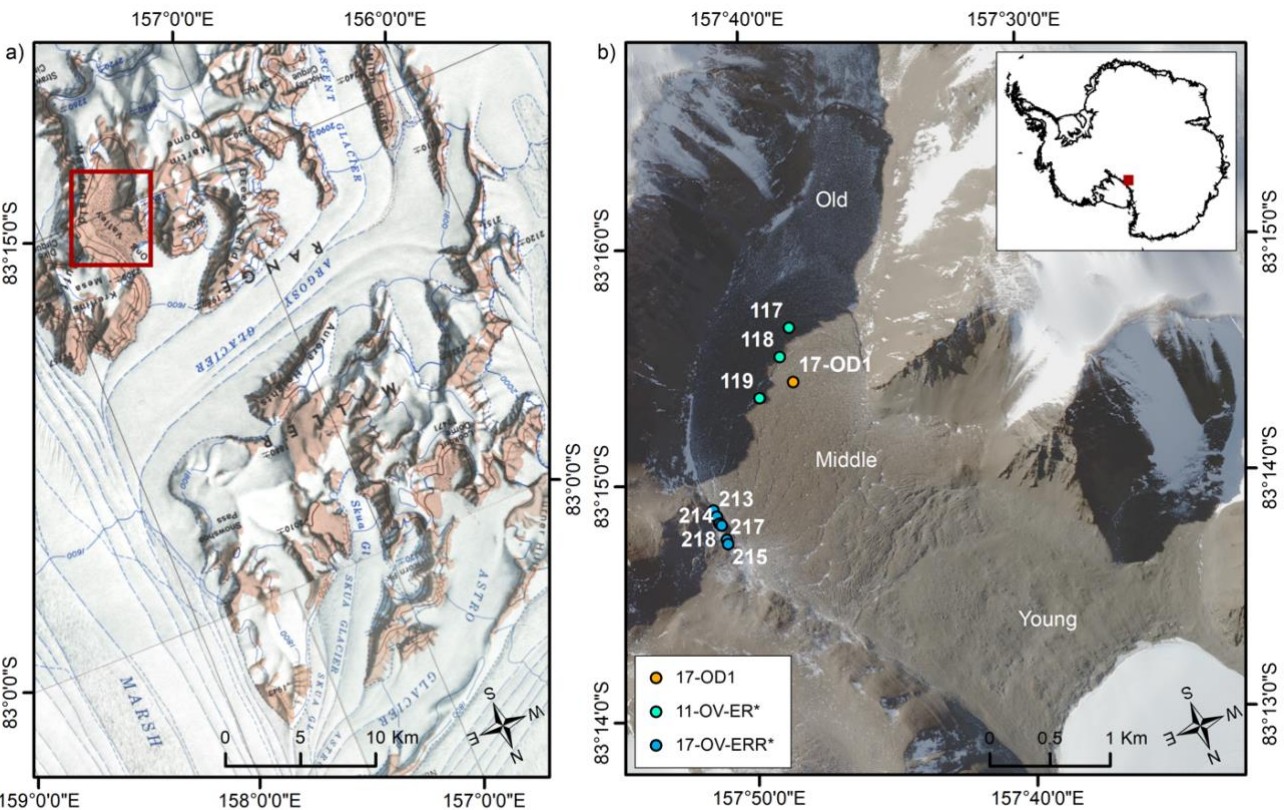

**Figure 1 Location of Ong Valley. (a) Cropped USGS 1:250,000 scale topographic map of Miller Range, Antarctica, showing the location of Ong Valley. The red rectangle indicates the location of Ong Valley opening perpendicular to Argosy glacier. (b)**
**Worldview 2 satellite image of Ong Valley, Antarctica (© 2016 Maxar). The dots indicate sampling sites for pit and ice core (orange), middle drift surface boulders (cyan), and lateral moraine boulders (blue). The legend shows the prefixes of sample names.**

The valley floor is mostly covered by a well-developed and distinctive system of three glacial drifts; referred to as young,

middle, and old (Bibby et al., 2016) (Fig. 1 and 2). These deposits were first described in 1975 and later identified as soil

chronosequences, increasing in age and maturity with distance from the Argosy glacier (Mayewski, 1975; Scarrow et al.,

2014). Bibby et al. (2016) found that the three drift units were ablation tills formed by sublimation of debris-rich glacier ice

that advanced into the valley. Eventually the ice became stagnant and began to sublimate, which led the englacial debris to

accumulate on the surface as supraglacial debris. Although some of the supraglacial debris in Ong Valley could originate

from a rockfall or colluvium from adjacent slopes, the drifts either have convex topography (middle and younger drifts) or

are bounded by prominent moraine ridges (older drift), and therefore significant input from local slopes is only possible

immediately adjacent to valley walls. In addition, surfaces of active glaciers in the region uniformly lack significant surface



sediment. While aeolian sediment transport onto the drifts is possible, drift surfaces are mainly composed of clasts and boulders too large for aeolian transport.

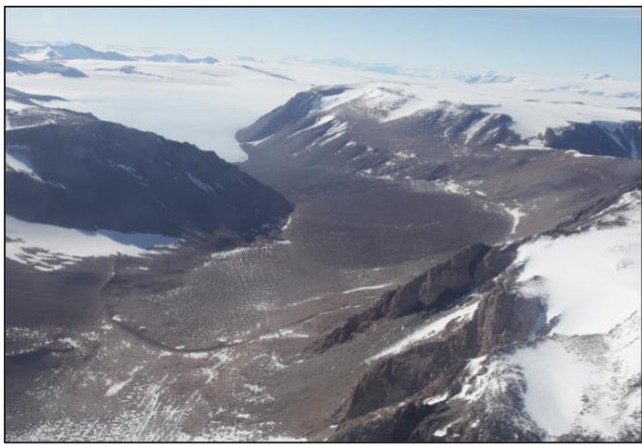
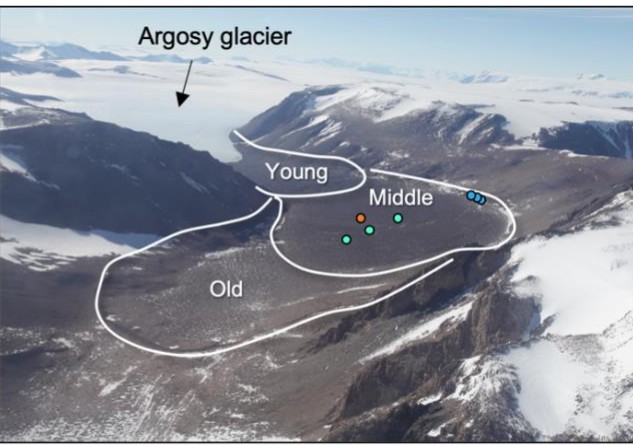

**Figure 2 Oblique aerial photograph of Ong Valley (left) with added markings (right) indicating the young, middle, and old drifts (Bibby et al., 2016). The photograph is looking northward and down valley. The dots represent the sampling sites as identified in Fig. 1.**

Exposure dating of the supraglacial debris from each drift has revealed the ages of 11-13kyr (young), >1.1 Myr (middle), and >2.7 Myr (old)(Bibby et al., 2016). The young and middle drift have buried ice under 0.1-0.5 m (young) and 0.6-0.8 m 95 (middle) of loose supraglacial debris completely concealing the ice. In this paper we refer to the buried ice below the middle drift as the middle ice. Contrary, the oldest drift is devoid of buried ice, which, presumably, has sublimated over extended exposure (Bibby et al., 2016). The highest surface elevation of these buried ice masses in the valley is located on top of the middle drift and ~200 m above the current Argosy glacier surface elevation at the valley mouth. Bibby et al. (2016) used cosmogenic-nuclide data from the surface debris layer to estimate that sublimation rates of 19-23 m Myr$^{-1}$ and surface 100 erosion rates of 0.7-0.9 m Myr$^{-1}$ have persisted in the valley since deposition of the drifts. All three drifts have related lateral moraines on the valley walls that trace the original elevation of the ice surface. The oldest drift also has a distinct end moraine close to the head of the valley, which shows no signs of influence from the small alpine glacier currently located at the head of the valley.

Major and minor mineral analysis of englacial debris and the supraglacial debris from each of the three drifts display shared provenance that are significantly different from the local bedrock of Ong valley (Edwards et al., 2014; Morgan et al., 2020). The local bedrock is primarily dominated by Hope Granite and the Argosy Gneiss (Barrett et al., 1970). This indicates that these drifts were deposited by past advances of the Argosy glacier.



## 3 Cosmogenic-nuclide applications relevant for dating Ong Valley buried ice

We use measured concentrations of cosmic-ray-produced nuclides in both the supraglacial debris covering the buried ice mass in Ong Valley and in debris within the buried ice in order to determine the age of the ice, its sublimation and erosion rate, and the geologic history of englacial debris. Cosmogenic nuclides are rare isotopes produced by cosmic-ray interactions with matter at Earth's surface (Dunai, 2010). Because the cosmic-ray flux is rapidly attenuated with depth below the surface, the concentration of cosmogenic nuclides can be used to date or quantify geologic processes that form or bury surface

materials (Lal, 1991; Nishiizumi et al., 1993). For example, nuclide concentrations in rocks or sediment brought rapidly from depth to the surface and not subsequently disturbed can be interpreted as an "exposure age" for rock or sediment. If such material is then re-buried, the decay of cosmogenic radionuclides can be used to compute a "burial age". In general, given a set of assumptions derived from the geologic context of a sample, measured cosmogenic nuclide concentrations can yield information about the timing of geologic events or the rates of geologic processes such as erosion and sedimentation that

have affected that sample.

There are three means by which we can apply cosmogenic-nuclide data to determine the age of the buried ice in Ong Valley. First, we can apply exposure-dating to the supraglacial debris; this approach was taken by Bibby et al. (2016). However, because the supraglacial debris is formed by sublimation after the deposition of glacial ice, the duration of the surficial

exposure of the supraglacial debris is, by definition, less than the age of the ice. Thus, in this case, a surficial exposure-dating is expected to yield only a minimum age. We can exposure date the supraglacial debris by determining the apparent exposure age of the surface, defined as the exposure age calculated from a nuclide concentration with the assumption of a single period of exposure continuing until today without erosion or burial. The basic assumptions for such exposure dating are that a sample (i) have never been exposed to cosmic-rays prior to entrainment; (ii) have only been exposed to cosmic-rays since

deposition; and (iii) have then never been covered, displaced, nor disturbed while exposed at the surface. Further, we can use two nuclides to quantify both erosion and exposure time (Lal, 1991). The concentration of each nuclide is a function of exposure age and surface erosion rate. Therefore, measurements of both nuclides yield two independent equations in which both unknows can be solved for in certain circumstances.

The second approach that we use to determine the age of the buried ice mass is generally referred to as "depth-profile dating" (Hidy et al., 2010), and involves measurements of both surface and subsurface nuclide concentrations. This approach relies on the observation that surface concentrations show a greater dependence on sublimation and/or surface erosion rates and lesser dependence on emplacement age compared to concentrations in the subsurface below several meters depth  (Stone et al., 1998; Braucher et al., 2009). Thus, paired subsurface and surface measurements can, in principle, yield a unique solution



for both ice age and erosion rate. In Ong Valley, we apply this approach to both the supraglacial debris and subsurface englacial debris in core by creating a forward model that predicts nuclide concentrations at all depths as a function of the age of a deposit, the surface erosion rate of the deposit, and, in this case, the ice sublimation rate leading to formation of the supraglacial debris deposit. Fitting this forward model to a data set then yields best-fitting estimates of these input parameters.


The third approach can be used if any of the englacial debris has formerly been exposed at the Earth's surface and subsequently buried. Then we can apply a burial dating method based on the decay of cosmogenic radionuclides produced during the initial period of exposure. The principle of burial dating is that different cosmogenic nuclides, such as $^{10}$Be and $^{26}$Al, in the same mineral are produced at a fixed ratio that depends on the production rates. Samples that have experienced a
single period of exposure at the surface have nuclide concentration ratios corresponding to the production ratio. If the two nuclides have different half-lives, burial of the sample to a depth at which the cosmic ray flux is diminished causes the observed ratio to change through time due to the different rates of radioactive decay (Lal, 1991). Thus, given several other assumptions, the ratio reflects the duration of burial. Although we had no prior reason to expect the englacial debris to be sourced from recycled surface debris, we show later that it is, in fact, the case in Ong Valley. Therefore, at our field site,
burial dating can be used as an approach to constrain the age of the ice.

## 4 Methods

### 4.1 Sample collection

During the Austral summer, 2017/2018, we collected; (i) pit samples from unconsolidated supraglacial debris at drill site 17-OD1, (ii) an ice core taken directly below the pit samples, and (iii) erratic boulders from other locations on the middle drift
surface and correlative lateral moraines (Fig. 1). The drill site OD1 was located at a central highpoint within the middle drift. We chose this site because any deformation of the buried ice should be minimized at this location, and colluvium and rockfall from the valley walls cannot reach the site. We determined the location and elevation of the core site using postprocessed differential GPS. Boulder samples were located using uncorrected handheld GPS and their elevations were checked against the Reference Elevation Model of Antarctica (REMA) Digital Elevation Model (DEM) (Howat et al., 2019).
Topographic shielding calculations for the sites follow Balco et al. (2008, with accompanying online material).



### 4.1.1 Vertical pit sampling of surficial regolith

We excavated a hand dug pit for sampling the vertical section of the supraglacial debris (Fig. 3a). The pit was located in the center of a patterned ground polygon formation in which the surface did not show signs of reworking caused by former active polygon boundaries. The supraglacial debris is a sandy diamict with clasts of all sizes up to large boulders. The debris

surface is covered by a lag deposit of clasts larger than approximately 5-10 cm. The clasts are mostly angular with occasional faceted and/or weakly polished surfaces. Clast lithologies include both local bedrock and other rock types not locally present. The supraglacial debris shows no sign of stratification nor presence of ice cemented regolith and a sharp boundary can be observed between the debris and the underlying debris-rich ice mass (Fig. 3a).

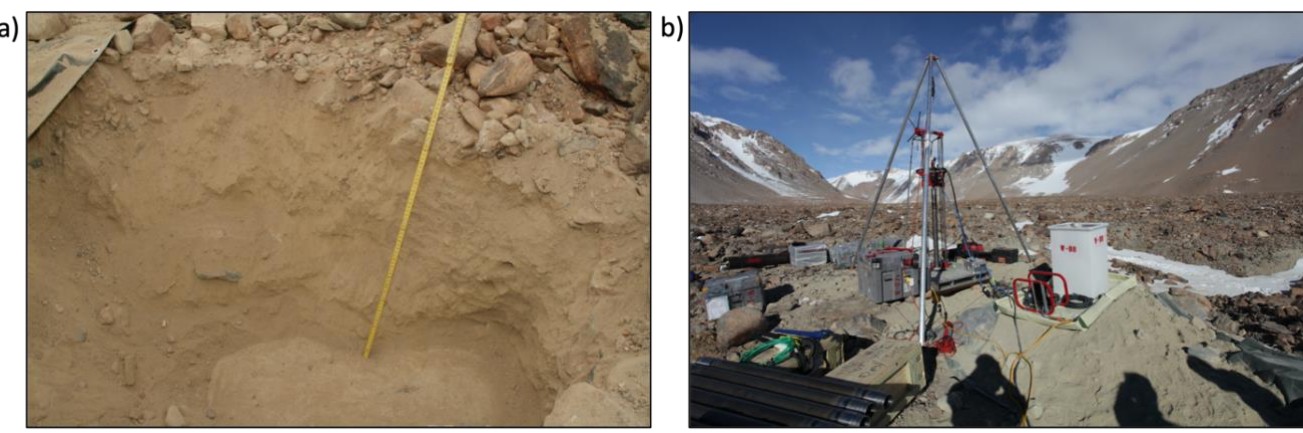

**Figure 3 Photographs of the drill site 17-OD1. (a) excavated pit for sampling of the supraglacial debris. The middle drift ice surface is found at the bottom of the pit. The yellow ruler in the image measure ~60 cm from the bottom of the pit to the surface of the supraglacial debris. (b) Winkie drill installed above the excavated pit for ice coring. Photograph looking south.**

Bulk sediment samples spanning 3-8 cm in sample depth were collected at approximately 10 cm depth intervals (Table 1). A total of 6 samples were collected throughout the pit, from the surface to the ice/debris boundary. The depth of the pit is 62

cm reaching from the surface of the supraglacial debris down to the ice surface.

### 4.1.2 Ice core sampling

We collected a 944 cm long and 7 cm diameter vertical ice core directly below the deepest supraglacial debris sample. This allows us to construct a depth profile of cosmogenic nuclide concentrations extending from the surface of the supraglacial debris to the bottom of the ice core. An updated Winkie drill (Fig. 3b) was used for drilling in mixed-media ice (debris-rich

ice) (Boeckmann et al., 2020). The recovered ice core was divided into segment lengths of 21 cm or less and stored frozen in individual watertight sample containers at the drill site. The samples were kept frozen at <-20°C until processed in the geochemistry laboratory at University of North Dakota.



### 4.1.3 Sampling of middle drift boulders

Samples of glacially transported boulders were collected during the 2011/12 field season (middle drift boulders) and the
2017/18 field season (lateral moraine boulders) (Fig. 4). Boulders were selected for sampling based on their evidence for
stability in the drift. We preferentially selected boulders that were partially buried in supraglacial debris, and that otherwise
showed no signs of overturning due to cryoturbation processes post-deposition. We also selected boulders that rose > 50 cm
from the debris surface to limit burial by snow. For the middle drift samples (sample prefixes 11-OV-ER*, Fig. 1), we
identified samples in the center of the valley to avoid any rockfall from the valley wall. For the lateral moraine boulders
(sample prefixes 17-OV-ERR*, Fig. 1) on the valley wall, we avoided rectangular boulders that were perched on the debris
surface as these appeared to be jointed boulders more recently eroded out of the bedrock. At all sites, deeply weathered
boulders were avoided in an attempt to limit samples with complex exposure histories. All boulders sampled were either
granite or gneiss that contained a high percentage of quartz. With a hammer and chisel, we removed a 1-2 kg sample from
the top of each boulder. GPS location, exposure geometry, and sample thickness were recorded. Boulders in the middle of
the drift should provide minimum exposure ages for the middle drift as they were exposed as the middle ice has sublimated.

**Figure 4 Photographs of surface boulders sampled from (a-c) the middle drift surface and (d-f) the East lateral moraine; (a) 11-OV-ER-117, (b) 11-OV-ER-118, (c) 11-OV-ER-119, (d) 17-OV-ERR-213, (e) 17-OV-ERR-217, and (f) 17-OV-ERR-218.**





## 4.2 Sample processing

### 4.2.1 Sample preparation

Each ice core segment containing visible englacial debris was weighed before the ice was melted at room temperature. The liquid was then separated from the englacial debris, and the debris was oven dried at ~70°C. Then the samples were re-weighed. Resulting masses of ice and debris in each core segment were then used to compute the density of the core using Eq. 4. Figure 5 shows pictures of the core and corresponding debris concentrations (blue) by weight as the function of depth. The ice core alternates between sections of clean ice and debris rich ice with a maximum debris concentration of 0.57 by weight. Ice core segments with little or no debris were assumed to have the density of ice, 0.917 g cm$^{-3}$.



**Figure 5** Image of the 17-OD1 ice core. The full image is stitched together from multiple individual pictures each covering approximately 20 cm of core length. The blue graph shows the corresponding debris concentration calculated by weight at depth below the surface. The topmost 62 cm is the thickness of the overlying supraglacial debris layer (not shown in pictures).



The density of the supraglacial debris was measured by packing the sediment sample into a measuring cup of known volume
and weighing it. This process was repeated five times for each sample to obtain a representative mean density of the
supraglacial debris of ~1.8 g cm$^{-3}$, excluding grains > 2 cm.

For cosmogenic nuclide analyses, the pit samples from the supraglacial debris were sieved to separate a grain size fraction of
250-500 mm, and the englacial debris was sieved to a grain size of 250-833 mm. This was done to maximize the amount of
sample for cosmogenic nuclide analyses. Quartz grains were isolated and cleaned following the procedure described in Stone
(2004). When needed, adjacent segments were merged so that enough quartz was available to permit precise beryllium (Be)
and aluminum (Al) measurements (see Table 1 for details).

### 4.2.2 Cosmogenic nuclide extraction and analysis

After quartz preparation but prior to dissolution for Be-Al extraction, a ~0.5 g aliquot of the prepared quartz was split off for
$^{21}$Ne measurements. These employed the "Ohio" noble gas mass spectrometer and extraction line at the Berkeley
Geochronology Center (BGC). Details of neon (Ne) isotope measurements on this system are described in Balco and Shuster
(2009) and Balter-Kennedy et al. (2020). $^{21}$Ne concentrations in replicate analyses of the CRONUS-A intercomparison
standard (Vermeesch et al., 2015) measured during analytical sessions in this study ranged from $314.3 \pm 9.4$ to $320.8 \pm 6.1$
Matoms g$^{-1}$, indistinguishable from the accepted value of 320 Matoms g$^{-1}$. In Tables 1 and 2, and in the supplement, we
report $^{21}$Ne concentrations as excess $^{21}$Ne relative to atmospheric composition. Excess $^{21}$Ne includes both cosmogenic $^{21}$Ne
and, potentially, nucleogenic $^{21}$Ne derived from uranium (U) and thorium (Th) decay. U and Th concentrations in quartz
from Ong Valley lithologies (Sams, 2016), expected Ne closure ages of these lithologies, and the observation that $^{21}$Ne
concentrations in drift and boulder samples from the youngest drift are significantly higher than expected for the last glacial
maximum (LGM) age of that drift, all indicate that nucleogenic $^{21}$Ne is significant in quartz in these lithologies. For samples
from supraglacial and englacial debris used to fit our forward model for nuclide accumulation, we did not make a correction
for nucleogenic $^{21}$Ne because it would be equivalent to inherited cosmogenic $^{21}$Ne in our model simulations. Thus, such a
correction would not affect values for ages or process rates inferred from the model simulations. In calculating exposure ages
and erosion rates for boulder samples, we corrected for nucleogenic $^{21}$Ne using an estimate of $7 \pm 3$ Matoms g$^{-1}$ obtained
from $^{21}$Ne data on boulders of similar lithology from the youngest drift (Sams, 2016). With this estimate, nucleogenic $^{21}$Ne
comprises 2-7% of total excess $^{21}$Ne in boulder samples. In addition, as discussed below, we applied this correction to
subsurface samples used for burial dating.



Chemical extraction and preparation of beryllium and aluminum from remaining quartz extracted from drill core samples were performed at the University of Vermont/National Science Foundation Community Cosmogenic Facility following the process described in Corbett et al. (2016). The pit and ice core samples were processed in two separate batches of 12 samples, in which each batch included a process blank and a standard. For all samples, 250 μg $^9$Be was added with a beryl carrier made at the facility with a concentration of 291 μg mL$^{-1}$. In addition, a $^{27}$Al carrier commercially available as an ICP standard from SPEX with a concentration of 1000 μg mL$^{-1}$ was added only to samples having < 1500 μg, of total Al. The amount of $^{27}$Al carrier added was based on the total amount of native $^{27}$Al in a sample quantified by Inductively Coupled Plasma Optical Emission Spectroscopy (ICP-OES) analysis. Quartz isolation and beryllium extraction for boulder samples followed the same procedure, except that three of the boulder samples (11-OV-ER-117, 118, 119) were processed in chemistry laboratories at the Center for Accelerator Mass Spectrometry, Lawrence Livermore National Laboratory (LLNL-CAMS).

Ratios of $^{10}$Be/$^9$Be measured at Lawrence Livermore National Laboratory (LLNL) are normalized to the 07KNSTD3110 standard (Nishiizumi et al., 2007) with an assumed $^{10}$Be/$^9$Be ratio of 2.85 x 10$^{-12}$. All $^{26}$Al/$^{27}$Al ratios were measured at PRIME and normalized to the KNSTD-01-5-2 standard (Nishiizumi, 2004) with an assumed $^{26}$Al/$^{27}$Al ratio of 1.818 x 10$^{-12}$.

Both $^{10}$Be and $^{26}$Al measurements were corrected for background using a procedural blank measured in each batch. Procedural blanks run with samples from the core site and were 14.9 ± 3.3 x 10$^3$ and 49.8 ± 9.4 x 10$^3$ atoms $^{10}$Be and 259 ± 45 x 10$^3$ and 37 ± 20 x 10$^3$ atoms $^{26}$Al, and blanks run with boulder samples ranged from 24 - 109 x 10$^3$ atoms $^{10}$Be. For the majority of samples, blank corrections account for less than 1% of total $^{10}$Be or $^{26}$Al atoms present. The exception is several samples of englacial debris from the ice core, for which blank corrections were up to 4% of total $^{26}$Al atoms present and up to 9% of total $^{10}$Be atoms present. The reported uncertainty in the measured nuclide concentrations accounts for all sources of analytical errors, including AMS measurement uncertainties, concentration measurement of $^{10}$Be and $^{26}$Al, and procedural blanks. Measurement details appear in the Supplement.

### 4.3 Forward Exposure model

The exposure history of the middle drift in Ong Valley is complex, and the nuclide production cannot be accounted for by simply exposure-dating the supraglacial debris. Therefore, we apply a forward model which attempts to account for the geological processes that result in the thickening of the supraglacial debris and accumulation of cosmogenic nuclides at depth. As described in Bibby et al. (2016) , the concentration of cosmogenic nuclides in the ice mass and the supraglacial debris is expected to result from a series of events: i) debris rich glacial ice was deposited into Ong Valley during glacial



advancement, ii) the ice mass became stagnant and began to sublimate which caused the englacial debris to accumulate on the ice surface as a supraglacial debris layer; iii) as the ice continued to sublimate additional debris was added to the

supraglacial debris layer from below, bringing deeper samples closer to the surface; iv) at the same time the supraglacial debris layer was subjected to surface erosion at a rate slower than the accumulation of debris from sublimation, such that the supraglacial debris thickness increased with time. The present-day thickness of the supraglacial debris layer is therefore a function of the age of ice emplacement, rate of ice sublimation, concentration of debris in ice, and rate of surface erosion. The numerical forward model attempts to account for the series of events listed above which lead to the cosmogenic nuclide

concentrations measured today at depth below the surface in the middle ice. Input parameters to the model include the age of ice emplacement, the sublimation rate of the ice, and the surface erosion rate of the supraglacial debris layer. The model then predicts nuclide concentrations in supraglacial and englacial debris. By fitting the model to the observed nuclide concentrations, we obtain estimates for the age of the ice and for sublimation and erosion rates.

### 4.3.1 Shielding Mass

The production rate of cosmogenic nuclides at and below the surface is dependent on the shielding mass which attenuates the cosmic ray flux. The shielding mass is the cumulative mass of sediment and ice overlying each sample per unit area and has units of g cm$^{-2}$. It is equal to the product of the sample depth (cm) and the mean density of the overlying material (g cm$^{-3}$). The density is related to the concentration of the suspended debris $C_D$ and the ice $C_I$ in each core segment, and is calculated from the total segment weight, $M_t$ (g), and dried sediment weight, $M_s$ (g) such that,

$$C_D = \frac{M_s}{M_t} \tag{1}$$

$$C_I = 1 - C_D \tag{2}$$

The density of the ice core can then be calculated by mixing the two ice core components based on volume. Assuming the density of the ice $\rho_I$ to be 0.917 g cm$^{-3}$ and the density of the debris $\rho_D$ to be that of rock, 2.68 g cm$^{-3}$, for an ice core segment weight of 1 g, the total weight of the ice core segment is then,

$$M_t = C_D + C_I \tag{3}$$

resulting in the density of the ice-debris mixture in an ice core segment $\rho_M$ to be,

$$\rho_M = \frac{1}{\left(\frac{C_D}{\rho_D}\right) + \left(\frac{C_I}{\rho_I}\right)} \tag{4}$$

### 4.3.2 Depth as a function of time

Once an ice mass with a mixture of ice and debris is emplaced at some time in the past defined as T (yr), the shielding mass

above a given sample in the ice is decreasing through time at the rate determined by the sum of the sublimation rate of the

—





ice and the surface erosion rate of the supraglacial debris. As the sample reaches the top of the ice, it becomes part of the supraglacial debris and then approaches the surface solely at the rate of surface erosion. The sublimation rate, s (cm yr$^{-1}$) is defined as a constant rate in which the surface of the ice-debris mixture (bottom of the supraglacial debris layer) is lowering. Note that this parameter represents a surface lowering rate due to sublimation and is not the same as a sublimation rate of

pure ice as would be considered in a thermodynamics context. Therefore, the initial surface of the ice mass is $sT$ (cm) above the present surface.

The rate at which mass is being lost by sublimation is the product of the sublimation rate and the density of the sublimating material. Since the ice mass consists of a mixture of ice and debris, only part of the ice mass is sublimating. The rate of mass

loss associated with sublimation is given by $s(1 - C_D)\rho_M$ (g cm$^{-2}$ yr$^{-1}$). While the ice is sublimating, the debris suspended in the ice mass is left behind on the ice surface and accumulating as supraglacial debris. The rate at which mass is added to the bottom of the supraglacial debris by sublimation is then, $sC_D\rho_M$ (g cm$^{-2}$ yr$^{-1}$). By assuming that at time of emplacement, the thickness of the supraglacial debris above the ice mass was zero, then with constant sublimation rate and erosion rate the total mass thickness of the supraglacial debris, $Z_T$ (g cm$^{-2}$) created by ice sublimation can be expressed as

$$Z_T = TsC_D\rho_M - TE_T \tag{5}$$

Where T is the age in years before present the ice was emplaced, and $E_T$ is the erosion rate expressed in mass units (g cm$^{-2}$ yr$^{-1}$). Equation (5) leads to the constraint that $sC_D\rho_M > E_T$, as the thickness of the supraglacial debris cannot be negative.

From field measurements, the thickness of the supraglacial debris is known to be 110 g cm$^{-2}$ for 17-OD1-Pit2. Therefore, for

any arbitrary values of age, sublimation rate, and erosion rate, the debris concentration must be chosen such that the measured supraglacial debris mass thickness is obtained after $T$ years. Assuming the ice mass mixture only consists of ice and debris, then the term $C_D$ and $\rho_M$ are different representations of the debris mass embedded in the sublimating ice.

Multiplying Eq. (4) with C$_D$, the debris concentration of the lost mass associated with sublimation can be solved,

$$C_D = \frac{\rho_D(C_D\rho_M)}{\rho_D\rho_I - (\rho_I - \rho_D)(C_D\rho_M)} \tag{6}$$

Equation (5) and (6) are two independent equations including the term $C_D\rho_M$. By isolating and substituting the term $C_D\rho_M$ in Eq. (5) into Eq. (6) the debris concentration now becomes independent of density of the ice core, and instead a function of sublimation rate ($s$), erosion rate ($E_T$) and age of ice emplacement ($T$),

$$C_D = \frac{\rho_D\left(\frac{Z_T + TE_T}{Ts}\right)}{\rho_D\rho_I - (\rho_I - \rho_D)\left(\frac{Z_T + TE_T}{Ts}\right)} \tag{7}$$



The debris concentration is constrained such that $0 \geq C_D \leq 1$. Further, Eq. (5) leads to the constraint that $sC_D\rho_M > E_T$, as the thickness of the supraglacial debris cannot be negative.

When predicting the concentration of cosmogenic nuclides, it is crucial to know a sample's depth at present, defined as $Z_{S,now}$ in units of mass depth (g cm$^{-2}$), and its depth at some time, t (yr) in the past, Z(t). Since the samples collected consist of both

ice core samples and sediment from the above laying supraglacial debris, there are two separate cases of how a sample has approached the surface in the past.

In case 1, the sample is in the ice at present such that the sample depth is greater than the depth of the supraglacial debris, $Z_{s,now} > Z_{till}$. From time of ice emplacement, the sample has then approached the surface at the rate of the ice sublimating and

the rate of surface erosion, such that

$$z(t) = Z_{S,now} + ts(1 - C_D)\rho_M + tE_T \qquad (8)$$

In case 2, the sample is in the supraglacial debris at present, such that $Z_{s,now} < Z_{till}$. The time that the sample has been in the supraglacial debris is then defined as the mass height of the sample above the supraglacial debris base depth divided by the

rate of mass addition to the supraglacial debris,

$$T_{till} = \frac{(Z_T - Z_{S,now})}{sC_D\rho_M} \qquad (9)$$

By the time of emplacement, the sample has then approached the surface in the same way as above. However, once the sample reaches the top of the ice and becomes part of the supraglacial debris, it approaches the surface at the rate of surface erosion only. This further allows for two scenarios to occur.


In case 2a, if at time $t$ the sample is in the supraglacial debris, such that $t < T_{till}$, then

$$z(t) = Z_{S,now} + tE_T \qquad (10)$$

In case 2b, if at time $t$, the sample is in the ice, where $t > T_{till}$, then

$$z(t) = Z_{S,now} + T_{till}E_T + (t - T_{till})(s(1 - C_D)\rho_M + E_T) \qquad (11)$$

With this, it is now possible to calculate the depth of a sample at any given time in the past since the ice emplacement. This is needed in order to predict the total accumulation of nuclides in a sample, which is dependent on the nuclide production at depth.





### 4.3.3 Cosmogenic nuclide production at depth

The cosmogenic nuclides [10]Be, [26]Al, and [21]Ne are produced by high-energy spallation, negative muon capture, and fast muon interactions (Dunai, 2010). The high-energy spallation particles are likely to react with mass in the atmosphere and at Earth's surface. Therefore, the production of cosmogenic nuclides due to spallation reaction is highest at the surface and considerably decreases with depth. Muons are much less likely to interact with mass, and therefore travel farther below the surface before stopping (Lal, 1991). While cosmogenic nuclide production at depth below the surface is solely due to muon production, muons are responsible for less than 1% of the total production at the surface for all nuclides (Balco, 2017).

We calculated the [10]Be production rate using the 'LSDn' scaling method (Lifton et al., 2014) as implemented in version 3 of the online exposure age calculator originally described by Balco et al. (2008) and subsequently updated, and the CRONUS-Earth "primary" calibration data set (Borchers et al., 2016). This yields a long-term (> 1 Ma) average production rate due to spallation of 25.3 atoms [10]Be g$^{-1}$quartz yr$^{-1}$. We then assumed that the [21]Ne/[10]Be production ratio is 4.03 (Balco et al., 2019) and the [26]Al/[10]Be production ratio is 6.75 (Balco et al., 2008).

The spallation production rate at the surface $P_{sp}(0)$ for a given cosmogenic nuclide decreases exponentially with depth (Lal, 1991), such that

$$P_{sp}(z) = P_{sp}(0) \, e^{\left(\frac{-z}{\Lambda}\right)} \tag{12}$$

Where z is the mass depth (g cm$^{-2}$) and L is the attenuation length, defined as the distance where the energetic cosmic-ray flux intensity reduces by a factor of $e^{-1}$ due to scattering and absorption processes. The attenuation value varies depending on altitude and latitude and is taken to be 140 g cm$^{-2}$ in Antarctica for [10]Be, [26]Al, and [21]Ne (Borchers et al., 2016; Balco et al., 2019).

Muon production rates are not well approximated by a single exponential function. As depth increases, the energy of the remaining muons that have not yet stopped is higher, and therefore it takes proportionally longer for those to stop. The calculations of the production rates due to negative muon capture follows that of Heisinger et al. (2002a) and production rates due to fast muon interactions according to Heisinger et al. (2002b) and are combined into a total muon production rate at depth, $P_{\mu}(z)$. The surface topography surrounding the sampling site also shields the samples of cosmic rays and will need to be accounted for when computing the production rate. This topographic shielding scaling factor $S_G$ is applied to the spallation and not the muon production rate (Balco et al., 2008). A topographic shielding of 0.993 was measured for drill site 17-OD1. The total production rate as function of depth can then be described as,



$P(z) = S_G P_{sp}(z) + P_\mu(z)$  (13)

In Fig. 6 we show the calculated changes in mass depth and production rate for a sample collected in the supraglacial debris 50 cm below the surface. The following arbitrary, but illustrative model parameter values are used for ice emplacement age, sublimation rates, and erosion rates: 1 Ma, 20 m Myr$^{-1}$, and 0.1 m Myr$^{-1}$ respectively. The supraglacial debris thickness is

that measured at drill site 17-OD1 and is 62 cm (110.15 g cm$^{-2}$). From the time of ice emplacement, a sample's depth has decreased linearly due to ice sublimation and surface erosion as the age of the ice increases, with a distinct change in rate once the sample exits the ice and becomes part of the supraglacial debris following that of Eq. (10) and Eq. (11). It is also observed that such samples experience great changes in nuclide production rates, following that of Eq. (13), from time of emplacement until present.

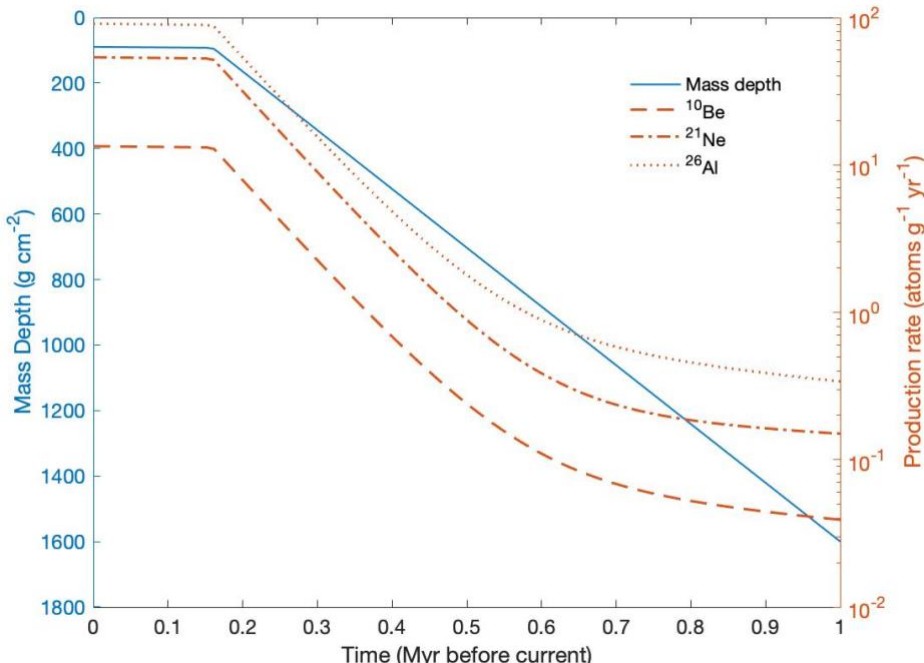


**Figure 6 Graphical representation of the temporal change in mass depth (blue line) and production rates (orange lines) since ice emplacement for a sample found today at 90 g cm$^{-2}$ (~50 cm below the surface) in the supraglacial debris. Here, the given sample has an initial shielding mass of 1600 g cm$^{-2}$ (~14 m) and approaches the surface at the combined rate of sublimation and erosion (Eq. 11). Once the sample reaches the ice surface it becomes part of the supraglacial debris (~0.2 Ma) and approaches the surface**

**solely at the rate of surface erosion (Eq. 10). As the sample's mass depth decreases it experiences considerable changes in production rates of cosmogenic nuclides (Eq. 13). The arbitrary model parameter values for these calculations are: 1 Ma ice emplacement, 20 m Myr$^{-1}$ sublimation rate, and 0.1 m Myr$^{-1}$ erosion rate.**





### 4.3.4 Predicting cosmogenic nuclide concentration at depth

When exposed to cosmic-rays, a sample will begin to accumulate cosmogenic nuclides over exposure time $T$, such that the
total accumulation, $N$ (atoms g$^{-1}$) in a subsurface sample at depth $z$ (g cm$^{-2}$) can be expressed as the integral of the production
rate a sample undergoes from time of ice emplacement to the present. Since the exposure history of the englacial debris goes
beyond the exposure history of the ice mass, some amount of inherited background nuclides $N_{inh}$ (atoms g$^{-1}$) are present.
While the concentration of the stable cosmogenic nuclide $^{21}$Ne continues to build up, some of the unstable radionuclides,
$^{10}$Be and $^{26}$Al, are lost to radioactive decay. This is expressed as an exponential, such that the total number of cosmogenic
radionuclides for a sample can be calculated using Eq. (14) and simplified to Eq. (15) for the stable nuclide $^{21}$Ne.

$$N_i = N_{i,inh}e^{(-\lambda_i T)} + \int_0^T P_i\big(z(t)\big)e^{(-\lambda_i t)}dt \tag{14}$$

$$N_{21} = N_{21,inh} + \int_0^T P_{21}\big(z(t)\big)dt \tag{15}$$

The subscript $i$ refers to the radionuclide of interest, $^{10}$Be or $^{26}$Al, and $\lambda$ is the decay constant for the radionuclide $i$. The
decay constants used in this paper are 4.99 x 10$^{-7}$ and 9.83 x 10$^{-7}$ for $^{10}$Be and $^{27}$Al, respectively. We evaluate these integrals
numerically using the default algorithm (integral) in MATLAB.

Given a set of environmental conditions; i) age of ice emplacement, ii) sublimation rate of ice, iii) surface erosion rate of the
supraglacial debris, and iv) inherited nuclide concentration for each of the nuclides, as well as measurable site conditions
(e.g., elevation and supraglacial debris thickness) Eq. (14) and Eq. (15) predicts the current total cosmogenic nuclide
concentration in a sample at a unique depth $z$ at present time.

### 4.3.5 Model fitting statistics

We defined a misfit statistic to compare observed nuclide concentrations with those predicted by the model as the reduced $\chi^2$
statistic weighted by the relative uncertainty in measurements of all three nuclide concentrations in each sample. A best fit is
found using the constrained nonlinear multivariable optimizing function (fmincon) in MATLAB while optimizing for the
free parameters; (i) the age that the ice was emplaced, (ii) sublimation rate of the ice since emplacement, (iii) surface erosion
rate of the accumulating supraglacial debris, and (iv) the inherited nuclide concentrations in the englacial debris at the time
of ice emplacement.

Uncertainty distributions on the best-fit values of the model parameters are derived from a 10,000-iteration Monte Carlo
simulation. Each Monte Carlo iteration draws from a set of normally distributed uncertainty for the measured nuclide
concentrations and uses an initial guess for the free model parameters based on published values for the middle drift in Ong



Valley (Bibby et al., 2016); 1.83 Ma ice emplacement age, 22.7 m Myr$^{-1}$ sublimation rate, 0.89 m Myr$^{-1}$ surface erosion rate, and inherited nuclide concentration of 0.14 x 10$^6$ atom g$_{qtz}$$^{-1}$, 11.4 x 10$^6$ atom g$_{qtz}$$^{-1}$, and 0.82 x 10$^6$ atom g$_{qtz}$$^{-1}$ for $^{10}$Be, $^{21}$Ne, and $^{26}$Al, respectively. Although uncertainties in calculated nuclide production rates are similar in magnitude to the uncertainty measured in nuclide concentrations, they are not included in these results as all the samples are from the same location, and therefore must have the same surface production rate. Further, any of the Monte Carlo simulation results which did neither converge nor satisfy the optimization function evaluation or resulted in an ice emplacement age younger than the last glacial maximum (LGM) (< 20ky) were excluded.

## 5. Results

### 5.1 Ice core visual observations

Visual inspection of the ice core indicate that it is primarily composed of debris-rich ice containing poorly sorted sediment ranging from clay to clasts exceeding the diameter of the borehole (Fig. 4). Bands of clean ice (lacking debris) were observed with a thickness < 0.5 m. Sections of the debris-rich ice show distinct debris bands with a thickness of a few centimeters. The orientation of these debris bands is variable but commonly steeply dipping.

### 5.2 Measured Ice Core Nuclide Concentrations

We measured cosmogenic nuclide $^{10}$Be, $^{21}$Ne, and $^{26}$Al in 6 supraglacial debris pit samples (17-OD1-PIT2*) and 12 ice core samples (17-OD1-C1*). In addition, $^{21}$Ne was measured for the surface sample (17-OD1-surf). The results show that cosmogenic nuclide concentrations in the englacial debris show large downcore variations (Table 1). The relative magnitude of the variation is larger for $^{10}$Be and $^{21}$Ne than for $^{26}$Al, which indicates that the amount of time that has elapsed since ice emplacement is most likely on the order of several half-lives of $^{26}$Al (the $^{26}$Al half-life is 0.7 Ma).



**Table 1 Measured cosmogenic nuclide concentrations in quartz extracted from supraglacial debris (prefix 17-OD1-surf/PIT2\*) and ice core (prefix 17-OD1-C1\*)**

| Sample ID | Top depth (cm) | bottom depth (cm) | Top Mass Depth (g cm⁻²) | Bottom Mass Depth (g cm⁻²) | $^{10}$Be x $10^6$ (atoms g$^{-1}_{qtz}$) | $^{21}$Ne x $10^6$ (atoms g$^{-1}_{qtz}$) | $^{26}$Al x $10^6$ (atoms g$^{-1}_{qtz}$) |
|---|---|---|---|---|---|---|---|
| 17-OD1-surf | 0 | 1 | | | | $130.6 \pm 7.3$ | |
| 17-OD1-PIT2-6-14 | 6 | 14 | 11.4 | 25.8 | $13.431 \pm 0.084$ | $109.1 \pm 6.2$ | $67.8 \pm 1.2$ |
| 17-OD1-PIT2-14-17 | 14 | 17 | 25.8 | 31.1 | $13.512 \pm 0.085$ | $102.1 \pm 4.6$ | $67.3 \pm 1.4$ |
| 17-OD1-PIT2-23-29 | 23 | 29 | 41.8 | 52.5 | $12.866 \pm 0.080$ | $104.9 \pm 5.8$ | $63.4 \pm 1.3$ |
| 17-OD1-PIT2-37-43 | 37 | 43 | 66.7 | 77.4 | $11.593 \pm 0.070$ | $91.7 \pm 4.3$ | $58.2 \pm 1.2$ |
| 17-OD1-PIT2-43-50 | 43 | 50 | 77.4 | 89.6 | $6.639 \pm 0.059$ | $55.2 \pm 2.8$ | $35.59 \pm 0.74$ |
| 17-OD1-PIT2-56-62 | 56 | 62 | 99.9 | 110.2 | $4.246 \pm 0.052$ | $34.6 \pm 2.5$ | $23.32 \pm 0.53$ |
| 17-OD1-C1-5-36 | 67 | 98 | 114.7 | 145.4 | $0.709 \pm 0.020$ | $11.4 \pm 1.5$ | $4.43 \pm 0.21$ |
| 17-OD1-C1-36-48 | 98 | 110 | 145.4 | 158.2 | $0.573 \pm 0.014$ | $12.2 \pm 1.1$ | $3.44 \pm 0.15$ |
| 17-OD1-C1-48-70 | 110 | 132 | 158.2 | 180.5 | $0.573 \pm 0.013$ | $13.3 \pm 2.9$ | $2.97 \pm 0.15$ |
| 17-OD1-C1-70-100 | 132 | 162 | 180.5 | 216.9 | $1.459 \pm 0.030$ | $39.9 \pm 1.1$ | $4.22 \pm 0.22$ |
| 17-OD1-C1-107-125 | 169 | 187 | 223.3 | 248.3 | $1.096 \pm 0.010$ | $37.3 \pm 5.8$ | $3.331 \pm 0.086$ |
| 17-OD1-C1-125-145 | 187 | 207 | 248.3 | 269.9 | $0.871 \pm 0.017$ | $30.4 \pm 2.5$ | $2.83 \pm 0.10$ |
| 17-OD1-C1-185-235 | 247 | 297 | 306.6 | 354.7 | $0.288 \pm 0.010$ | $14.0 \pm 2.8$ | $1.48 \pm 0.12$ |
| 17-OD1-C1-235-310 | 297 | 372 | 354.7 | 426.9 | $0.1603 \pm 0.0090$ | $6.4 \pm 1.4$ | $1.17 \pm 0.10$ |
| 17-OD1-C1-310-350 | 372 | 412 | 426.9 | 468.7 | $0.1161 \pm 0.0048$ | $9.2 \pm 5.7$ | $0.831 \pm 0.055$ |
| 17-OD1-C1-500-582 | 562 | 644 | 607.0 | 694.5 | $0.1479 \pm 0.0041$ | $21.2 \pm 1.0$ | $0.450 \pm 0.040$ |
| 17-OD1-C1-582-649 | 644 | 711 | 694.5 | 774.6 | $0.1323 \pm 0.0032$ | $28.3 \pm 7.4$ | $0.409 \pm 0.032$ |
| 17-OD1-C1-781-819 | 843 | 881 | 895.8 | 938.1 | $0.4184 \pm 0.0081$ | $52.6 \pm 4.1$ | $0.489 \pm 0.034$ |
| 17-OD1-C1-819-879 | 881 | 941 | 938.1 | 1001.2 | $0.516 \pm 0.010$ | $64.5 \pm 4.1$ | $0.49 \pm 0.031$ |
| 17-OD1-C1-879-944 | 941 | 1006 | 1001.2 | 1071.7 | $0.4816 \pm 0.0091$ | $67.0 \pm 2.4$ | $0.471 \pm 0.038$ |




### 5.3 Boulder surface exposure results

We measured cosmogenic nuclide [10]Be and [21]Ne in quartz from erratic boulders from the middle drift surface and east lateral moraine (Fig. 1). Apparent [10]Be and [21]Ne exposure ages for the middle drift surface boulder samples are 1.55-2.16 Myrs and 0.82-1.39 Myrs for the East lateral moraine boulders (Table 2). Further, the apparent [10]Be exposure ages reported for all

boulder samples appear younger than of those from [21]Ne, and is an indication that some process (e.g. erosion, burial, etc.) must have occurred which decreases the [10]Be nuclide concentration relative to [21]Ne





**Table 2 Exposure-age data for boulders on the surface of Ong Valley middle drift and correlative lateral moraines.**

| Sample name | Latitude (DD) | Longitude (DD) | Elevation (m) | Sample thickness (cm) | Shielding factor | [10Be] (Matoms g-1) | Apparent 10Be exposure age (Ma) | Excess [21Ne] (Matoms g-1) | Apparent 21Ne exposure age (Ma) |
|---|---|---|---|---|---|---|---|---|---|
| Boulders on drift surface in valley center | | | | | | | | | |
| 11-OV-ER-117 | -83.25658 | 157.70622 | 1601 | 2 | 0.9916 | $25.15 \pm 0.43$ | $1.546 \pm 0.039$ (0.143) | $206.2 \pm 7.0$ | $2.157 \pm 0.082$ (0.157) |
| 11-OV-ER-118 | -83.25478 | 157.71861 | 1610 | 3.5 | 0.9950 | $27.49 \pm 0.46$ | $1.778 \pm 0.048$ (0.176) | $191.7 \pm 5.8$ | $2.002 \pm 0.071$ (0.143) |
| (Replicate 21Ne measurement) | | | | | | | | $195.6 \pm 5.8$ | $2.044 \pm 0.071$ (0.145) |
| 11-OV-ER-119 | -83.25237 | 157.74124 | 1597 | 1.5 | 0.9929 | $28.28 \pm 0.45$ | $1.854 \pm 0.049$ (0.187) | $181.4 \pm 5.5$ | $1.885 \pm 0.068$ (0.135) |
| Boulders on lateral moraine correlative with middle drift on east side of valley | | | | | | | | | |
| 17-OV-212-ERR | -83.24582 | 157.7961 | 1715 | 1.5 | 0.9909 | $19.09 \pm 0.14$ | $0.9280 \pm 0.0088$ (0.070) | $120.8 \pm 5.3$ | $1.093 \pm 0.059$ (0.090) |
| 17-OV-213-ERR | -83.24529 | 157.79558 | 1718 | 2 | 0.9909 | $18.48 \pm 0.23$ | $0.892 \pm 0.014$ (0.068) | $106.3 \pm 4.6$ | $0.955 \pm 0.053$ (0.080) |
| 17-OV-214-ERR | -83.24472 | 157.79579 | 1714 | 6 | 0.9909 | $18.54 \pm 0.23$ | $0.936 \pm 0.015$ (0.072) | $129.0 \pm 6.7$ | $1.215 \pm 0.073$ (0.105) |
| 17-OV-215-ERR | -83.24336 | 157.79526 | 1733 | 2.5 | 0.9909 | $17.42 \pm 0.14$ | $0.8205 \pm 0.0080$ (0.060) | $99.1 \pm 3.9$ | $0.873 \pm 0.047$ (0.071) |
| 17-OV-217-ERR | -83.24301 | 157.79561 | 1715 | 3.5 | 0.9918 | $18.11 \pm 0.19$ | $0.885 \pm 0.012$ (0.067) | $138.3 \pm 5.2$ | $1.271 \pm 0.058$ (0.098) |
| 17-OV-218-ERR | -83.24453 | 157.79491 | 1722 | 1.5 | 0.9918 | $19.63 \pm 0.18$ | $0.954 \pm 0.011$ (0.073) | $154.2 \pm 5.8$ | $1.394 \pm 0.062$ (0.106) |

Notes:

1. Rock density is assumed to be 2.57 g cm-3 based on measurements on like lithologies in Ong Valley.

2. Exposure ages are calculated using default production rates and 'LSDn' scaling in version 3 of the online exposure age calculator described by Balco et al. (2008) and subsequently updated.

3. Both internal (including only measurement uncertainty) and external (in parentheses; also includes production rate uncertainty) uncertainties are shown for apparent exposure ages.

4. 21Ne exposure age calculations include subtraction of $7 \pm 3$ Matoms/g non-cosmogenic 21Ne (see text).



## 6 Model fitting

In the following sections we first highlight several important features of the data from the ice core and supraglacial debris that we seek to explain using the forward model for nuclide accumulation described above. We then fit the forward model to the observations and thereby obtain estimates for the emplacement age and sublimation rate of the buried ice. Lastly, we calculate a minimum age for the middle drift in Ong Valley.

### 6.1 Qualitative observations of the ice core data

The forward model described in sect. 4.3.4 is an exposure model that is based on an assumption that the englacial debris is well mixed and therefore all the samples contain the same amount of inherited nuclides. The model calculates the postdepositional nuclide production during the exposure (Eq. 14 and 15). This is compatible with an ice mass that is emplaced during single glacial advance. After a single event of exposure, where sublimation and erosion has occurred, the concentration for a given cosmogenic nuclide in the englacial debris must decrease monotonically with depth as the

production rate decreases with increased shielding mass.

However, multiple sections of the ice core show increase in cosmogenic nuclide concentration at depth (Table 1) and is therefore not compatible with a single exposure history described above. The most likely explanation for the observed increases in cosmogenic nuclide concentrations with depth in OD1 samples, is that englacial debris in various sections of the

core have variable exposure histories prior to entrainment in the ice, and therefore have different inherited nuclide concentrations. Based on the measured nuclide concentrations we make the following two observations that guide our forward modeling.

The first observation is that the set of samples that display monotonically decreasing nuclide concentrations (Fig. 7. Sections

S1, E1 and E2) are segments of the ice core with relatively low debris concentrations. These samples follow the expectation of having an exposure history as outlined in the forward model. We define these samples to be of a low-nuclide concentration that is largely composed of subglacially derived material sourced from upstream of the Argosy Glacier and have minimal surface exposure prior to entrainment. The debris from such samples is therefore identified to be of 'englacial debris' (samples denoted englacial E1 and E2 and highlighted in blue in Fig. 7) including the current surface debris layer

(S1) that we assume has also originated from englacial debris.





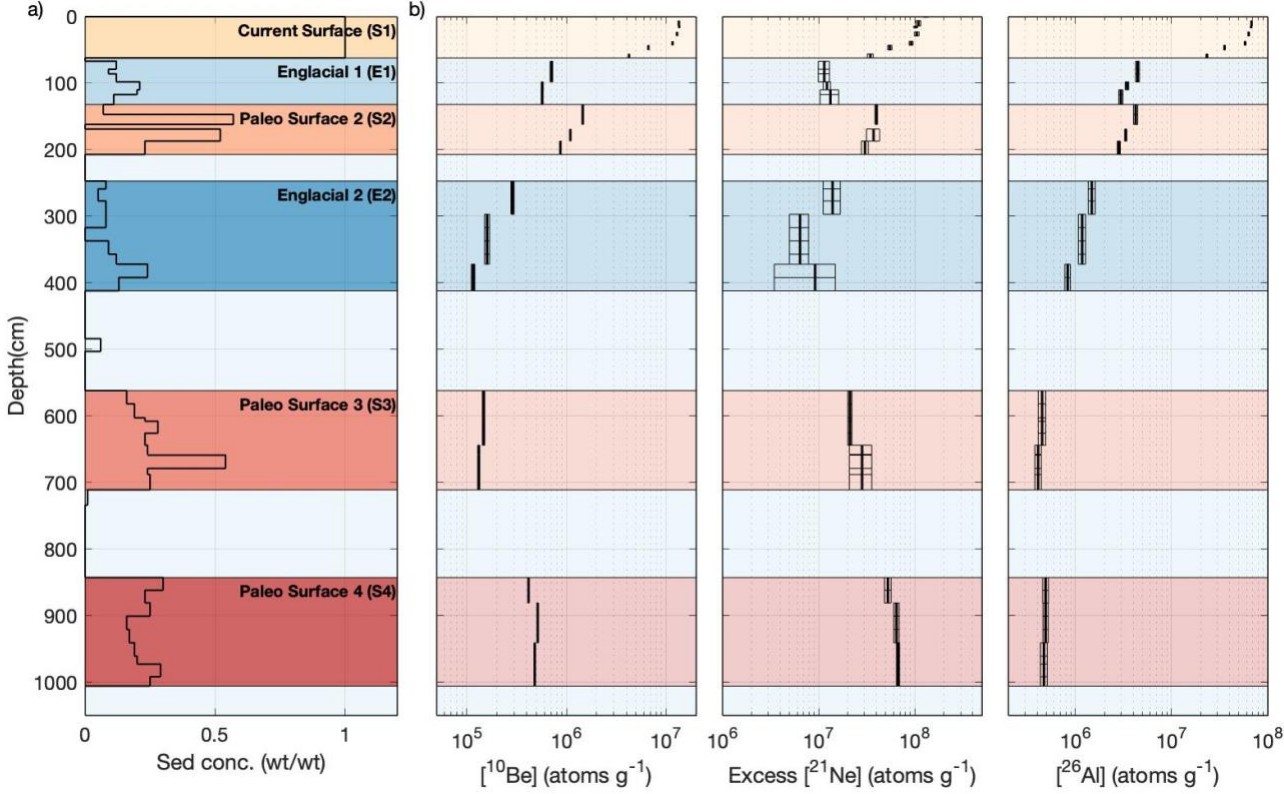

**Figure 7 Depth plot of (a) sediment concentration (weight sediment/weight total sample) and (b) measured ¹⁰Be, ²⁶Al, and ²¹Ne nuclide concentrations. Color shades highlight the source of the debris. Yellow are current supraglacial debris samples (S1) and has same origin as blue samples (E1 and E2) identified as 'englacial debris' having no prior surface exposure before being subglacially entrained. Samples in shades of red is identified as 'paleo surfaces' (S2, S3, and S4) and to be of recycled surface debris that was exposed on the surface and entrained as the glacier overrode previously ice-free areas as it advanced into Ong Valley.**

The second observation is that the debris-rich sections of the ice core (Fig. 7a. Sections S2, S3, and S4) have much higher nuclide concentrations and do not conform to the assumptions of a well-mixed ice mass containing subglacially derived debris and only experiencing *in situ* englacial accumulation of cosmogenic nuclides. We consider the debris in these sections to be from one or more high-concentration sources that are likely composed of material that was exposed for an extended period of time on the surface and entrained as the glacier overrode previously ice-free areas as it advanced into Ong Valley. In fact, these samples have ²¹Ne concentrations similar to modern surface material (S1), and therefore must have been exposed for millions of years at or near surface. Therefore, debris from these subsurface samples having higher nuclide concentrations are identified as 'paleo surfaces' and is observed in three distinct units. Although these units must all contain





some fraction of recycled surface debris, this may be of variable origin, so we identify these as S2, S3, and S4 (highlighted in shades of red in Fig. 7). In this paper we refer to this debris as recycled paleo surface.

Based on the above observations we can only explain a subset of the data with the forward model. However, the presence of
significant and variable inheritance in the debris from different sources suggests that we can apply a burial constraint to the inherited nuclide inventory having prior exposure. When fitting the model to the postdepositional nuclide inventory, this allows for an additional constraint on the age of ice emplacement.

## 6.2 Forward modeling used to explain the data set

The forward model predicts the accumulation of cosmogenic nuclides $^{10}$Be, $^{26}$Al, and $^{21}$Ne in the ice core and the overlaying supraglacial debris during a single event of exposure, constrained by following rates: sublimation of ice, surface erosion, and accumulation of supraglacial debris (Eq. 14 and Eq. 15). If we assume constant inheritance in the englacial debris, it is only possible to fit the model to the set of samples that show monotonically decreasing nuclide concentrations with depth. The model is therefore fitted to; i) surface sample in which only $^{21}$Ne measurement is available, ii) all measured nuclide samples
in the supraglacial debris, S1, and iii) samples of englacial debris, units E1 and E2.

While the recycled paleo surfaces S2, S3, and S4 are not included in the fitting of the forward model, they are utilized for burial-dating to further constrain the age of ice emplacement. The general idea for applying burial-dating to the recycled paleo surface samples is following. The high cosmogenic nuclide concentrations in the recycled paleo surfaces are the result
of extended period of exposure of the debris prior to entrainment in the middle ice. The debris in these samples was part of a surface that was overridden during the latest advance of glacial ice into Ong Valley. Hence, this paleo surface debris must have been buried at the time the middle ice was deposited. The burial age obtained from burial-dating of the debris from these recycled paleo surface samples, should then reflect the timing of the middle ice emplacement. Further, the burial age of these samples cannot display ages that are younger than the event at which they got buried. Therefore, the minimum burial
age for any of the recycled paleo surface sections (S2-4) serves as the maximum age for when the middle ice was emplaced in Ong Valley.

Burial-dating of the recycled paleo surfaces is applied to their inherited nuclide concentrations. The inherited nuclide concentration is calculated by subtracting the modeled post-emplacement nuclide concentrations from the total measured
nuclide concentrations. The apparent burial age of the recycled paleo surface debris is then determined from the nuclide ratio





of the calculated inherited nuclide concentrations. The apparent burial age is the duration of burial inferred from a pair of nuclide measurements under the assumption that a sample has experienced a single period of exposure followed by a single period of burial. In reality, the sample could have experienced multiple shorter periods of burial; however, the calculated apparent burial age is the maximum single period a sample has been buried for. The age of ice emplacement for the middle

ice is then limited by the minimum apparent burial age for any of the recycled paleo surface samples S2-4. This constraint is incorporated into the model fitting such that when adding the nuclide concentrations lost by decay during burial, the nuclide ratio does not exceed that of the surface exposure production ratio. This burial constraint is applied only to the samples that are not used for forward model fitting, and that make up the recycled surface material S2, S3, and S4.

## 6.3 Model Results

By fitting the forward model prediction to measured nuclide concentrations from the englacial debris sample (S1, E1 and E2) and applying burial constraint to sections of recycled paleo surface debris (S2, S3 and S4), we are able to constrain the age of ice emplacement, sublimation rate, and surface erosion rate for the Ong Valley middle ice. The results of a 10,000-iteration Monte Carlo simulation provide an ice emplacement age of 2.95 +0.18/-0.22 Myrs for the middle ice, with a best-fit, $\chi^2$ of 3.75 +0.98/-0.45. The best-fitting sublimation rate since emplacement is 22.86 +0.10/-0.09 m Myr$^{-1}$, with a surface erosion

rate of 0.206 +0.013/-0.017 m Myr$^{-1}$. The results of the simulation are not normally distributed, and the best-fit values are therefore reported as the 50$^{th}$ percentile with error bounds given by the 16$^{th}$ and 84$^{th}$ percentile (Fig. 8 and Supplementary Figure). The inherited nuclide concentration for $^{10}$Be, $^{21}$Ne, and $^{26}$Al (the initial nuclide concentration present in the ice mass at the time of deposition 2.95 Ma ago) is 0.101 +0.018/-0.017 x 10$^6$ atoms g$^{-1}$, 9.2 +1.4/-4.4 x 10$^6$ atoms g$^{-1}$, and 0.8249 +0.0062/-0.0031 x 10$^6$ atoms g$^{-1}$, respectively.





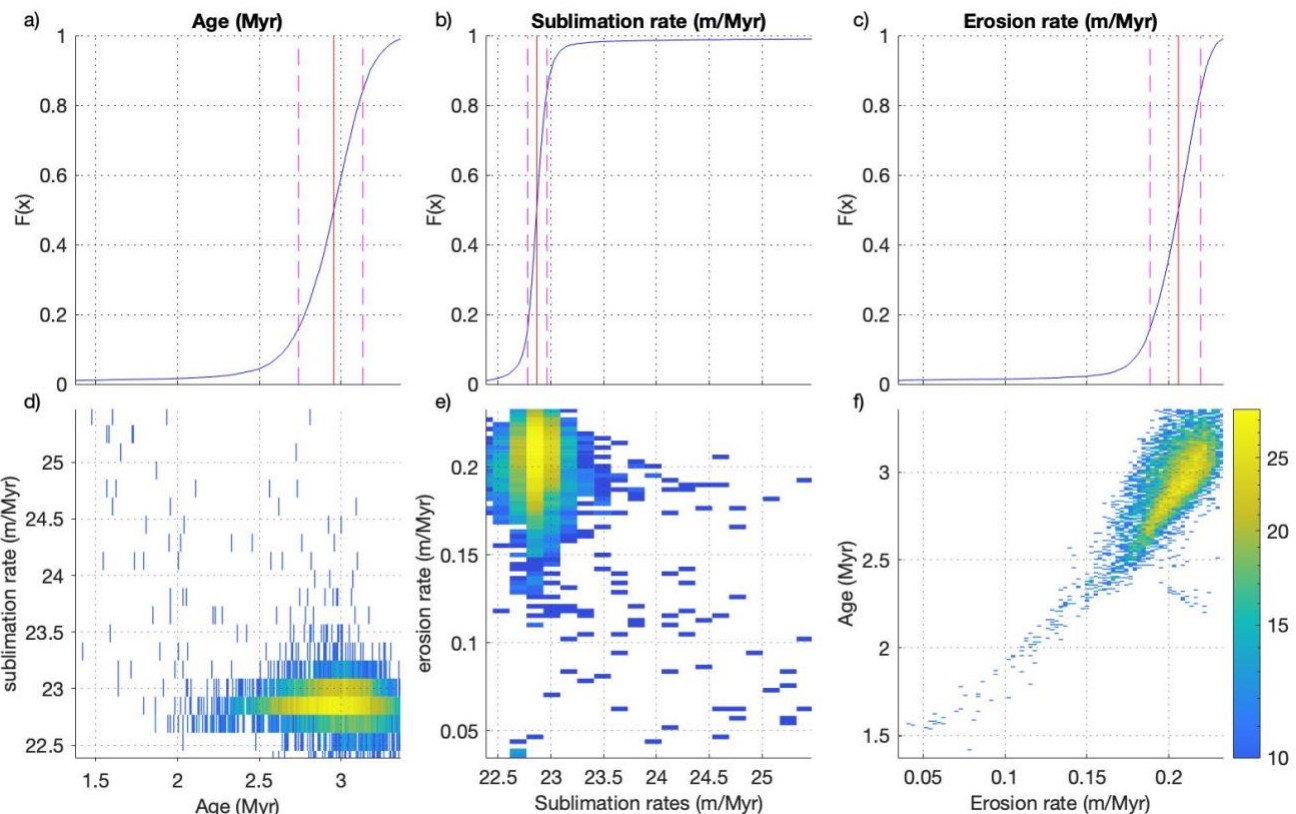

**Figure 8 (a-c) Cumulative distribution of 10,000 fitted Monte Carlo simulation results showing the 50th percentile (red line) with error bounds given by the 16th and 84th (pink lines). (f-e) Density plot as paired distribution of Monte Carlo simulation results separated into 1000 bins with yellow being high density and blue low density. Both the vertical and horizontal axes (d-f) are truncated to the 98% (0.01 and 0.99) confidence interval.**

The measured debris concentrations in the ice segments range between 0 for clean ice and 0.57 for debris-rich ice, with an average of 0.13 by weight for the core. The average debris concentration for the best fit in the sublimated ice which produced the supraglacial debris over a period of 2.95 Ma is 0.036-0.034 by weight, and therefore an average ice-debris density of ~0.931 g cm$^{-3}$. Further, the resulting sublimation rate of 22.86 m Myr$^{-1}$ over the span of 2.95 Ma, results in a total ice surface lowering of 67.6 m.

The model predicts nuclide concentrations at depth similar to those measured in the supraglacial debris (S1) and englacial debris (E1-2) and it is therefore evident that these units can be explained with the exposure model (Fig. 9a-c). Further, the paired-nuclide plot (Fig. 9d-f) clearly shows the distinction between the englacial debris data set (S1, E1-2) explainable by





the model and the paleo surface samples (S2-4) having high nuclide concentrations and low paired-nuclide ratio, hence,

different origin which require the addition of a complex burial and exposure history.

Conceptually, higher nuclide concentrations require long surface exposure and any disequilibrium in the paired-nuclide ratio (ratio below the steady-state-erosion zone) is the effect of burial after exposure (Lal, 1991). However, the presence of a significant inherited nuclide inventory could result in surface and subsurface samples having ratios below the production

ratio and therefore indicating an apparent burial age. In Fig. 9d-f this is observed as both the predicted and all measured nuclide concentration ratios fall within the burial zone, and not near the steady-state-erosion zone.

**Figure 9** Measured and modeled cosmogenic nuclide concentrations as (a-c) mass depth below the surface for (a) [10]Be, (b) [21]Ne, and (c) [26]Al, and (d-f) paired-nuclide exposure-burial diagram for (d) [26]Al-[10]Be, (e) [10]Be-[21]Ne, and (f) [26]Al-[21]Ne pairs. Blue lines indicate the model predicted cosmogenic nuclide concentration from the pit surface to the bottom of the ice core. Colored boxes (a-c) and sample colors (d-f) indicate debris source as detailed in Fig. 7. In (d-f), solid black lines show the steady-state erosion zone, and dashed lines show the burial lines as Million-year decay isochrons (See sect. 3 for more details). The measured nuclide concentration for each sample is represented by a shaded ellipse of its 1-sigma uncertainty. The black line connects the sample ellipses from the surface of the pit down to the bottom of the ice core. *Nuclide concentrations normalized to respective surface production rate.





### 6.4 Minimum exposure age

We find the absolute minimum exposure age of the middle ice to be 1.331 +0.020/-0.024 Ma, with a sublimation rate of 24.70 +0.71/-0.56 m Myr$^{-1}$. This age is derived from the minimum possible number of assumptions about the geologic history of the samples. For a surface sample, the apparent age is the calculated age from the measured nuclide concentration assuming a sample has experienced a single event of exposure, zero surface erosion, and no burial during that time period. Under such assumptions, a surface sample's apparent exposure age serves as the minimum exposure age. Therefore, the

minimum age for the ice emplacement is obtained using the assumption that the inherited nuclide concentration for $^{10}$Be, $^{21}$Ne, and $^{26}$Al is equal to the minimum concentrations measured throughout the core (0.12 x 10$^6$ atoms g$^{-1}$, 6.42 x 10$^6$ atoms g$^{-1}$, and 0.41 x 10$^6$ atoms g$^{-1}$, respectively) with zero surface erosion.

### 6.5 Burial dating of Paleo Surface Debris

As evident in Fig. 9, the paleo surface samples have elevated nuclide concentrations and do not fit our modeled predictions.

There is no scenario in which these samples can be explained solely by our forward exposure model which includes only sublimation and erosion. Therefore, these samples must have experienced significant periods of surface exposure prior to subglacial entrainment. Further, in order to have a lower paired-nuclide ratio than predicted (Fig. 9d-f), the samples must have experienced at least one period of burial. Hence, these observations were the reasons for the inclusion of burial-dating in our model.


Similar to the burial-dating constraint added to the forward model (sect. 6.2), we can determine the burial age of these paleo surface samples by first subtracting modeled postdepositional nuclide concentrations at the sample depths from the measured concentrations. This yields an estimate of the nuclide concentrations present in the paleo surface samples (S2-4) at the time they were buried, less the effect of subsequent radioactive decay. The choice to only fit the model to a subset of samples is

based on the assumption that the paleo surface samples have different geological history and thus different nuclide inheritance from that of the englacial debris (E1-2) samples. Therefore, the estimated inherited nuclide concentration for these paleo surfaces obtained from this subtraction is different from the inherited nuclide concentrations inferred from the model fitting. From these inherited nuclide concentrations in the paleo surface samples, we can then solve for the burial age which would cause a sample exposed at the surface (plotting on the simple exposure line) to have paired-nuclide ratios as

shown in Fig. 9. Uncertainties on the burial ages are derived from the same Monte Carlo simulation used to generate uncertainty estimates for the model parameters.





The burial ages for the three paired-nuclide ratios, $^{26}Al/^{10}Be$, $^{10}Be/^{21}Ne$, and $^{26}Al/^{21}Ne$ for S2 are 3.21 ± 0.20 Myr, 4.20 ± 0.27 Myr, and 3.69 ± 0.21 Myr, respectively. The paleo surface S3 and S4 indicate longer periods of burial, with S3 have burial age of 4.33 ± 1.00 Myr, 7.58 ± 0.61 Myr, and 6.24 ± 1.35 Myr, and S4 has burial ages of 5.06 ± 0.25 Myr, 6.61 ± 0.12 Myr, and 5.78 ± 0.15 Myr, respectively for each of the three nuclide pairs. Figure 10 shows the paired nuclide ratios for each of the paleo surfaces as their apparent burial ages.

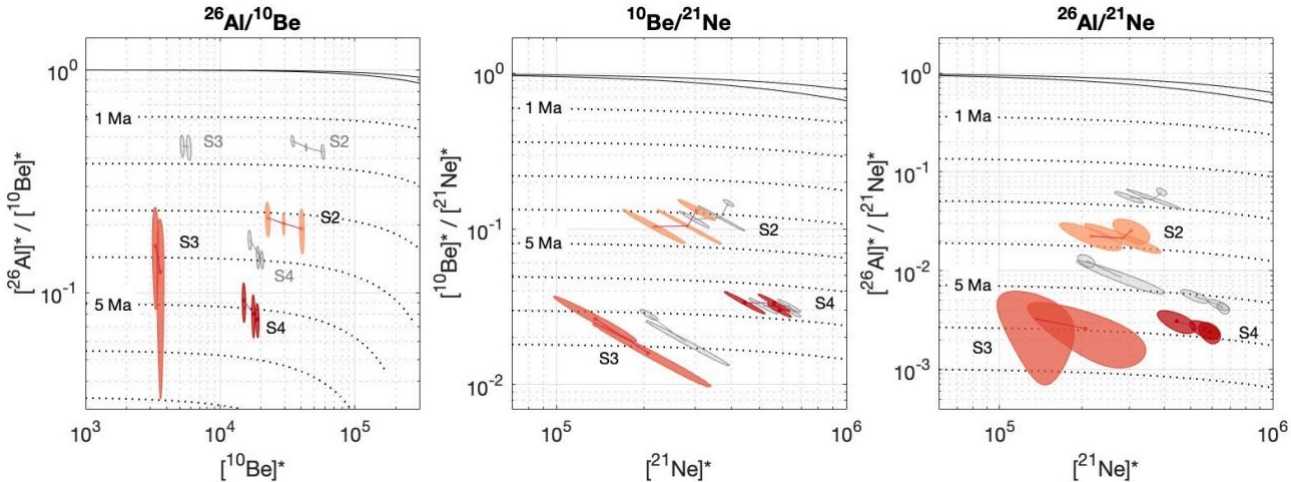

**Figure 10 Paired-nuclide diagram for (a) $^{26}Al$-$^{10}Be$, (b) $^{10}Be$-$^{21}Ne$, and (c) $^{26}Al$-$^{21}Ne$ pairs. Solid black lines show the steady-state erosion zone, and dashed lines show the burial lines as Million-year decay isochrons. The nuclide concentration for each data point is represented by a shaded ellipse of its 1-sigma uncertainty. Grey data points show the measured nuclide concentrations for the paleo surface samples as described in Fig. 9d-e. Shaded red data ellipses show the resulting nuclide concentrations when subtracting the modeled nuclide concentration from the measured. The burial age (dashed isochron lines) for which a sample lies represent the apparent burial age and is the maximum single period the sample has been buried since current time. Color shades refer to the different paleo surface units as described in Fig. 7. *Nuclide concentrations normalized to respective surface production rate.**

## 7. Discussion

### 7.1 Sublimation rate

The sublimation rate is tightly constrained between 22.77 – 22.96 m Myr$^{-1}$ (Fig. 8b) and independent of the age and erosion rate (Fig. 8d,e). With increasing sublimation rate, a sample having low nuclide concentration at deeper depth (caused by increased shielding mass) approaches the ice surface more rapidly. Having spent less time near the ice surface, a sample found in the top meter of the ice will have a much lower total nuclide concentration than a sample at the lower part of the supraglacial debris. This difference in nuclide concentration between the uppermost ice sample and the bottom supraglacial debris sample allows for the sublimation rate to be well constrained.




Previous estimates of the sublimation rate in Ong Valley range between 19-22 m Myr[-1] (Bibby et al., 2016), where sublimation rates of buried ice masses determined using cosmogenic nuclides elsewhere in TAM range between 0.7 - 37 m Myr[-1] (Schäfer et al., 2000; Ng et al., 2005; Morgan et al., 2010b). These rates broadly agree with a sublimation rate of 22.86 m Myr[-1] as reported here in Ong valley. Sublimation rates obtained from modeled water vapor diffusion are order of
magnitude higher; 100-500 m Myr[-1] (Hindmarsh et al., 1998; Mckay et al., 1998; Kowalewski et al., 2006; Hagedorn et al., 2007; Mckay, 2008; Schorghofer, 2009). Such orders of magnitude higher sublimation rates would suggest a total surface elevation lowering of ~ 300 – 1500 m as compared to our calculated total ice surface lowering of ~68 m and are inconsistent with glacial moraine elevations and field observations in Ong Valley.

The sublimation rate represents an average rate since the ice emplacement ~3 Myr ago. Most likely the sublimation has been decreasing over time as the supraglacial debris thickens (Mackay and Marchant, 2016). However, this relationship is uncertain, and therefore we do not account for it.

### 7.2 Erosion rate

The erosion rate and age of the ice are well constrained within an erosion-age tradeoff (Fig. 8f). For an eroding surface,
debris of low nuclide concentrations from below will approach the surface at a rate of erosion. With increased surface erosion rate, an older exposure age is required in order to account for the loss of high surficial nuclide concentrations, leading to an expected erosion-age tradeoff.

The majority of Antarctic studies of subaerial surface erosion rate using cosmogenic nuclides are obtained from boulders and
bedrock of various lithologies (Marrero et al., 2018, and references therein). Only a few erosion rates have been determined from surficial regolith (Putkonen et al., 2008; Morgan et al., 2010a; Bibby et al., 2016). While Bibby et al. (2016) found a 0.89 m Myr[-1] for the middle drift, a range between 0.2 – 2 m Myr[-1] has been observed in McMurdo Dry Valleys (Putkonen et al., 2008; Morgan et al., 2010a). Therefore, an erosion rate of 0.206 m Myr[-1] as reported here for the supraglacial debris is in agreement.

### 7.3 Englacial Debris Concentration

In Ong Valley, we measured an average debris concentration of 0.13 by weight in the ice (Eq. 1) which is in the same range as measurements made in Beacon Valley (0.085 by Marchant et al., 2002; and 0.Morgan et al., 2010a). While the modeled debris concentration of 0.035 by weight in the sublimated ice over a span of 2.95 Ma is lower than measured debris



concentration of buried ice in Antarctica, it is consistent with the expectation that the debris content increases towards the

bottom of glacial ice, due to subglacial entrainment of the debris. Thus, it is expected that the modeled debris concentration

for the sublimated ice here results in a lower concentration than measured in the remaining basal ice.

### 7.4 Mixing layer

Predicted cosmogenic nuclide concentrations in the supraglacial debris decreases with depth at a higher rate than measured

nuclide concentrations (Fig. 9a-c). This leads to a systematic misfit between observations and model predictions. By either

decreasing the sublimation rate, increasing the erosion rate, and/or decreasing the age of ice emplacement, a steeper

predicted cosmogenic nuclide depth profile can be obtained for the supraglacial debris. However, neither of these scenarios

will result in a better fit for the near-surface pit samples. The difficulty of fitting the forward model to the near-surface pit

samples suggests that partial vertical mixing of the supraglacial debris may have occurred.

As the supraglacial debris is accumulating due to sublimation, debris having low cosmogenic nuclide concentrations from

below will mix with debris of higher surficial cosmogenic nuclide concentrations. Therefore, any (partial or full) vertical

mixing of the supraglacial debris would cause a decrease in the cosmogenic nuclides inventory in any above lying sample.

Without accounting for any vertical mixing of the supraglacial debris, the model predictions result in an overestimation of

the cosmogenic nuclide concentration for the near-surface pit samples and an underestimation of newly accumulating

supraglacial debris from the ice surface as observed in Fig. 9a-c. Further, any vertical mixing would decrease the nuclide

ratio, which would explain why all paired-nuclide ratios for the supraglacial debris samples plots below the steady state

erosion zone (Fig. 9d-f).

It has previously been suggested that no vertical mixing occurs in the supraglacial debris layers in Ong valley (Bibby et al.,

2016) and supraglacial debris layers studied in Beacon Valley (Morgan et al., 2010a, b). While the current measured nuclide

profile may not reflect a fully mixed zone as seen elsewhere in temperate climate with bioturbation (Perg et al., 2002), a

partially mixed supraglacial debris layer is likely the result of active polygon formation found at the surface of the middle

drift.

### 7.5 Exposure ages from boulders

In general, a boulder having experienced a single period of exposure that is equal to the ice emplacement age of the middle

drift should display concordant [10]Be and [21]Ne ages. However, all measured boulders show apparent [10]Be exposure ages

younger than that of [21]Ne ages (Table 2) and are therefore inconsistent with a simple exposure having negligible erosion.



From the $^{10}$Be/$^{21}$Ne ratio (Fig. 11) it is observed that the surface boulders have experienced erosion while exposed to cosmic-
rays at the surface as the paired-nuclide ratio lays within the steady-state erosion zone (see details in sect. 3). Three outliers
of the east lateral moraine boulders (214, 217, and 218; Fig. 11a) show neither age or erosion rates that agree with
continuous exposure and lie below the steady erosion zone, in a region of intermittent exposure. Thus, these boulders show a
complex exposure history having experienced at least one period of burial at some point in time.

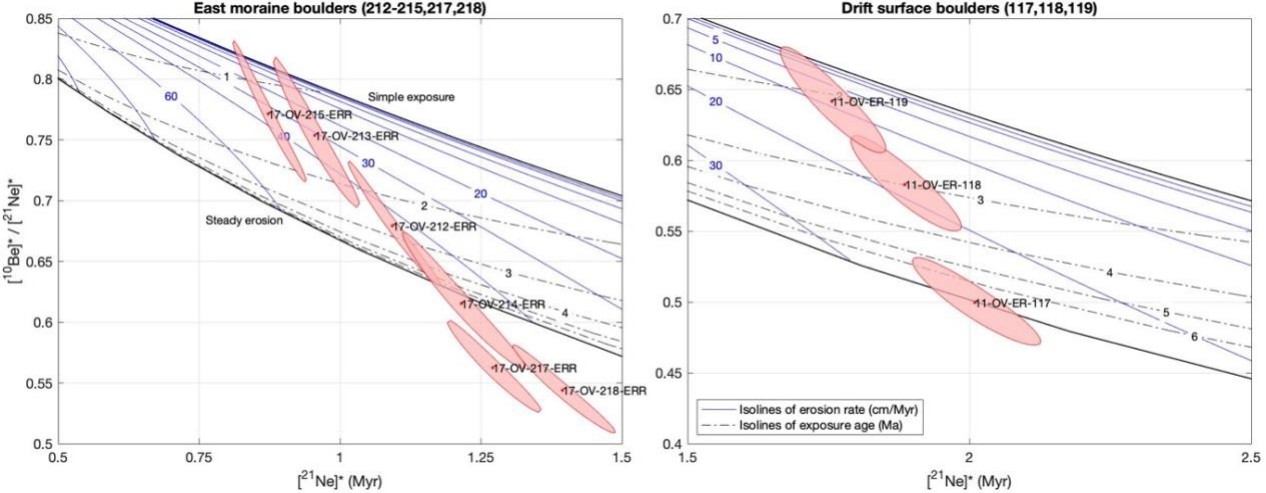

**Figure 11 $^{10}$Be - $^{21}$Ne paired-nuclide diagram of the boulder samples. Black solid lines are the simple exposure line and steady state erosion line, which marks the zone of continuous exposure. Blue lines are isolines of constant steady erosion (cm Myr-1), and black dashed lines are isoline of constant exposure age (Myr). The measured nuclide concentration for each sample is represented by a red dot with red shading of its 1-sigma uncertainty. *Nuclide concentrations normalized to respective surface production rate.**

A more realistic exposure age and erosion rate can be determined for boulders having a nuclide ratio within the steady-state
erosion zone. By assuming a single period of continuous exposure at a steady state erosion, we can solve for both the
exposure and erosion rate as detailed in Balco et al. (2014). The results of a 10,000-iteration Monte Carlo simulation using
this procedure are shown in Fig. 12. Some samples permit infinite ages at a steady erosion rate if the $^{10}$Be/$^{21}$Ne nuclide ratio
lies outside of the continuous exposure zone. Therefore, only samples permitting finite age-erosion rate solutions are shown
in Fig. 12.





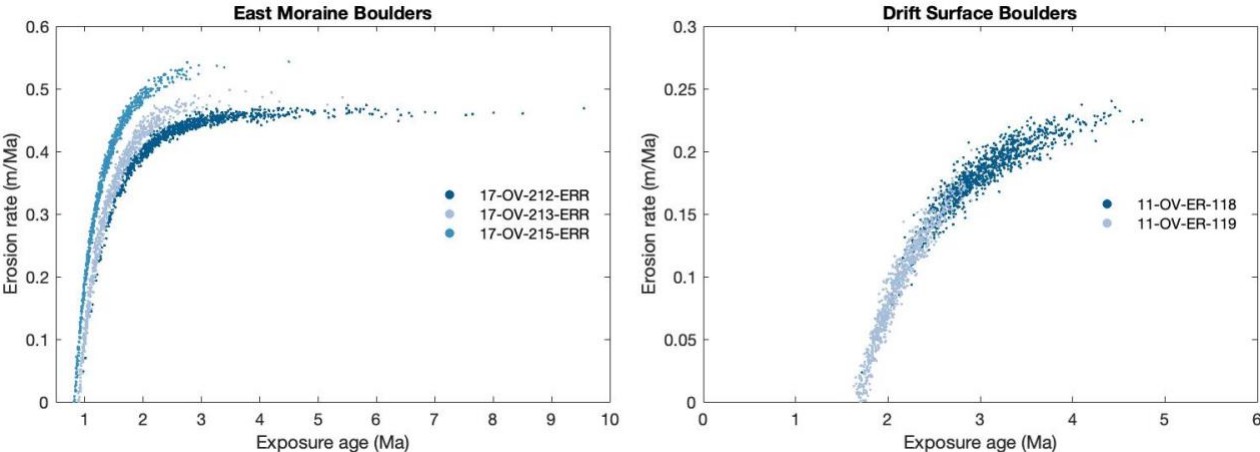


**Figure 12 Exposure ages and erosion rates for the middle drift surface and moraine boulders.**

The lateral moraine boulders having a finite age-erosion rate solution display a 68% confidence bound on the age of 1.0 – 3.9 Ma and erosion rates of 0.20 - 0.48 m Myr$^{-1}$ (Fig. 12a). From field observations, the lateral moraine from which boulder measurements were sampled appears to be a younger recessional moraine and therefore not an indication of a maximum

extent for the middle drift, which is observed at higher elevation. These observations would suggest that boulders from this lateral moraine have most likely been disturbed post ice emplacement. Thus, the $^{10}$Be-$^{21}$Ne ratio age for the moraine is more likely to represent a minimum limiting age of ice emplacement (Hallet and Putkonen, 1994; Putkonen and Swanson, 2003).

The middle drift surface boulders have a 68% confidence bound on the age of 1.8 - 3.5 Ma with erosion rates ranging

between 0.05 - 0.21 m Myr$^{-1}$ and therefore agree with our modeled age of 2.96 Ma for the middle drift ice (Fig. 12b). One outlier (11-OV-ER-117) has steady state erosion for a period greater than several half-lives of $^{10}$Be (> ~ 5 Ma), and therefore contains no age information. We attribute this increased nuclide concentrations to an extended period of exposure prior to deposition in Ong Valley.

**7.6 Multiple Glacial Events**

Samples having a burial age equal to that of the ice emplacement age are considered to have been derived from a paleo surface exposed in Ong Valley during ice advance of the middle ice. After entrainment into the advancing ice, the paleo surface material was immediately buried under a thick layer of ice and shielded from the cosmic-ray flux.



As mentioned in sect. 6.2, the age of the ice determined from modeled nuclide predictions is constrained such that the ice
cannot be older than the minimum burial age obtained from any sample across the three paired-nuclide ratios. We find that
the minimum burial age of 3.21 ± 0.20 Ma for S2 agrees with the age of the middle ice (2.95 + 0.18/-0.21 Ma). Hence, S2 is
likely to have been at the surface during the glacial advance leading to the deposition of the middle ice ~ 3 Ma. S2 is found
at depths in between E1 and E2 (Fig. 7), which have no prior exposure history. This would suggest that S2 is not in
stratigraphic order and has been mixed up into the ice during advance of the middle ice into Ong Valley.


Both S3 and S4 display older burial ages which are not uniform across the three paired-nuclide ratios (Fig. 10). This suggests
that both S3 and S4 have experienced a complex exposure-burial history of prior entrainment which goes beyond the
exposure history of the middle drift ice. The estimated burial ages represent the minimum total burial time a sample has
experienced, but also the maximum burial time of a single burial event. For S3 and S4, additional burial time is needed
beyond the age of the ice (> 2.95 Ma) and must have experienced multiple periods of burial. The simplest explanation is to
assume that during advance of the middle ice, Ong Valley looked similar to today, such that the middle ice advanced over an
already existing ice-cored drift unit. Perhaps this is now preserved as the older drift (Fig. 2). If this is the case, then S3 and
S4 units presumably are debris that was buried for some time in the older ice, and then buried again in or under the middle
ice. We use "in or under" because it is possible that (i) we cored through the middle ice into a stratigraphically underlying
mass of older ice. However, it is also possible that (ii) sections of the older ice were entrained during advancement of the
middle ice, and we then have a mixture of older and younger ice. The core did not display an obvious stratigraphic boundary.
In fact, most of the observed grain in the ice-debris mass is at steep angles and disturbed (Section 4.1.2), which would tend
to favor a mixing of the ice hypothesis. Regardless, either is possible and geochemical analysis of the ice could potentially
help resolve this.


The uncertainty associated with the age of this older, separate, underlying ice mass is greater compared to the middle ice due
to the complexity associated with the exposure-burial dating. That is, the burial age obtained here is the apparent burial of a
single event. However, a sample could have experienced multiple shorter periods of exposure-burial events which are not
accounted for. We therefore only report an estimate with minimum and maximum constraints. A sample that experiences a
single period of exposure at the surface have nuclide concentration ratio dependent on the duration of exposure. When buried
to a depth where nuclide production is significantly reduced, the change in ratio is primarily dependent on the radioactive
decay associated with the duration of burial. Therefore, when solving for the burial age for each of these paleosurface units,
we can also solve for the exposure age that has occurred prior to burial by the middle ice (Balco and Rovey, 2008).



With S2 representing a surface sample from a supraglacial debris prior to deposition of the middle ice, we find the surface exposure age of S2 to be $0.163 \pm 0.058$ Ma, $0.268 \pm 0.046$ Ma, and $0.268 \pm 0.046$ Ma for the paired-nuclide ratios, $^{26}Al/^{10}Be$, $^{10}Be/^{21}Ne$, and $^{26}Al/^{21}Ne$ respectively. This suggests that the age of an underlying ice mass is at least that of the minimum burial age plus the exposure age for S2. On the contrary, the age of the ice in which a sample is embedded cannot be older than the burial age of a given sample. Then, the maximum burial age across the three paleo surfaces S2-4 for any of the

paired-nuclide ratios must serve as the upper bound for the age. Therefore, there is no scenario in which this separate, older underlying ice mass can be younger than 3.3 Myr or older than 7.6 Myr. However, a more likely age for this older deposit would be 4.3 – 5.1 Ma which is the burial age obtained from the $^{26}Al/^{10}Be$ ratio for S3 and S4, as it is unclear whether or not S3 and S4 are the same or different units.

It is difficult to determine whether or not there is any defined boundary between the older and younger ice masses. With an increase of nuclide concentrations downcore and, in addition, that samples from E2, S3, and S4 appear to form mixing arrays in the two-nuclide diagrams shown in Fig. 9, it appears likely that S3 is a mixture of a high-nuclide-concentration end member, which may be represented by S4, and a low-nuclide-concentration end member represented by E2. However, a boundary or transition most likely exists between E2 and S4.

**7.7 Antarctica during the Pliocene**

The ages reported here coincide with the Pliocene Epoch (5.3 – 2.6 Ma). Research on Pliocene climate and how it affected the Antarctic ice sheet has gained much attention as a likely analog for modern anthropogenic warming (Dolan et al., 2018). During the Pliocene epoch there is evidence of prominent glacial deposits, in which two are identified in the southern hemisphere as globally recognizable glaciations (summarized in De Schepper et al., 2014); one occurring during the early

Pliocene (ca. 4.9-4.8 Ma), and another during the Late Pliocene (ca. 3.3 Ma), also identified as the Marine Isotope Stage (MIS) M2 glaciation. The latter is followed by a warmer-than present mid-Piacenzian Warming Period (mPWP; ~3.3-3.0 Ma)(Haywood et al., 2013; De Schepper et al., 2014; Dowsett et al., 2016). This warming period ends by a Late Pliocene cooling, post ~ 3 Ma, leading to a global glaciation around the Pliocene-Pleistocene boundary (De Schepper et al., 2014).

Because the uncertainty of the ice emplacement age ($\pm 0.2$ Ma) exceeds both the 40 kyr and 100 kyr climate cycles of the Pliocene epoch (Caballero-Gill et al., 2019), we are not able to accurately relate the deposition of the middle ice to an individual glacial event. Furthermore, the age of 2.95 +0.18/-0.22 Ma for the middle ice emplacement, which requires an East Antarctic Ice Sheet (EAIS) elevation greater than 200 m above present, cannot confidently be assigned to either the warmer period prior to 3 Ma or the cooler period post 3 Ma. Balter-Kennedy et al. (2020) concluded that glacial deposits

recording a higher than present EAIS elevation at Roberts Massif, a nearby location in the Transantarctic Mountains, most likely postdated the mPWP. Therefore, if the ice advance in Ong Valley were correlative with that at Roberts Massif, it would also be associated with the 3 Ma cooling. However, this is speculative.

The oldest englacial debris that we have dated in Ong Valley is dated at ~4.3-5.1 Myrs old. Although the dated age range is
rather wide due to complexities resulting from old age and exposure-burial dating, it is still direct evidence of an EAIS expansion and local ice expansion during that time. This dated age suggests that the ice sheet expansion predated the MIS M2 cooling event and possibly coincided with the Early Pliocene global glaciation (ca. 4.9 - 4.8 Ma). If in fact the older ice is still present below the middle ice mass, then it did not melt during a period of warming. Thus, additional evidence indicating whether or not two ice units are present would be important in understanding the climate during the Pliocene
Epoch. Since ~3 Myrs ago there has never been a comparable ice sheet expansion in Ong Valley as was seen in early/mid and late Pliocene. The only notable, but small ice sheet advance or stagnation evident in Ong Valley is the youngest drift dated at 11-13 kyrs.

## 8 Conclusions

Glacial ice is a well-known paleo climate archive. Great efforts have been made to find ice older than 1 Ma since the paucity
of ice beyond million years of age creates uncertainties for future climate predictions. In Ong Valley, Antarctica, the middle drift harbors a large ice mass buried 62 cm below the surface of supraglacial debris. We collected a 944 cm long ice core and measured concentrations of the cosmic-ray produced nuclides $^{10}$Be, $^{26}$Al, and $^{21}$Ne from the englacial debris and samples from the supraglacial debris directly above it. We developed a numerical forward model which predicts the accumulation of cosmogenic nuclides in the englacial debris and the above laying supraglacial debris during a single event of exposure,
constrained by sublimation, surface erosion and accumulation of supraglacial debris. The modeled nuclide concentrations are then fitted to the measured nuclide concentrations in the ice core.

Downcore increase in measured nuclide concentrations suggest that sections of englacial debris consist of both subglacially entrained debris and recycled paleo surfaces having a complex exposure-burial history prior to entrainment. This allows us to
apply a combination of exposure- and burial-dating to the forward model. We find the age of the middle drift ice mass to be 2.95 Ma, with a constant ice sublimation rate of 22.86 m Myr$^{-1}$ and surface erosion rate of 0.206 m Myr$^{-1}$. Cosmogenic nuclide exposure dating of surface boulders belonging to the surface of the coring site are consistent with the modeled age of ~3 Ma for the ice emplacement.

Exposure-burial dating on the englacial paleo surface debris reveals that the lower section of the ice core belongs to a separate and older deposit, emplaced ~4.3 – 5.1 Myr ago. We interpret this lower section as a portion of an older ice mass either *in situ* or transported during emplacement of the younger ice. The ages of the two separate ice masses found below the middle drift can be directly related to glacial advances. These findings provide direct evidence of an Antarctic ice sheet that was larger than present during the early and late Pliocene epoch.


Furthermore, we show that exposure-burial dating of cosmogenic nuclides measured *in situ* in basal ice debris layers can be used for age constraint of past ice advance. Specifically, we have debris layers in one ice core that suggest three different burial ages, where at least two of them are dated to be older than the age of the ice itself. This is important for understanding *in situ* cosmogenic-nuclide data from out of context subglacial sediment.


Collectively our results show that the continental ice sheet advanced into Ong Valley repeatedly and evidence of at least two of such advances at 2.95 Ma, and 4.3-5.1 Ma are still preserved in lateral moraines, drifts, and stacked ice masses. Since 2.95 Myrs ago the only evidence of ice advance or stagnation in the Ong Valley was ~10 kyrs ago.

**Code availability**

MATLAB code used for forward modeling and model fitting calculations is available at https://github.com/balcs/ong-valley-OD1-model.

**Data availability**

All data described in the paper are included in the Supplement.

**Author contributions**

MB, JP, GB, and DM conducted field work and sample collection. MB, LBC, GB, and DM carried out cosmogenic nuclide measurements and were responsible for data reduction. GB and MB developed the model for analysis. MB prepared the manuscript with contributions from all authors.



## Competing interests

The authors declare that they have no conflict of interest.

## Acknowledgements

This work would not have been possible without the efforts of many U.S. Antarctic Program personnel at McMurdo Station, Shackleton Glacier Camp, and elsewhere, as well as pilots and ground crew of the New York Air National Guard, Kenn Borek Air, and PHI, Inc. Development, deployment, and operation of the Winkie Drill was made possible by the US Ice Drilling Program, in particular Grant Boeckmann. Andrew Grant assisted with field work in Ong Valley, and Alan Hidy of
the Center for Accelerator Mass Spectrometry, Lawrence Livermore National Laboratory, assisted with AMS isotope ratio measurements and beryllium extraction chemistry. Satellite imagery in figure 1b was provided by the Polar Geospatial Center under NSF-OPP awards 1445205, 1445168, and 1445169.

## Financial Support

This project was supported by the U.S. National Science Foundation via grants OPP- 1445205, OPP- 1445168, and OPP-
1445169. In addition, Balco's work on this project was partially supported by the Ann and Gordon Getty Foundation. The University of Vermont Community Cosmogenic Facility is supported by the U.S. National Science Foundation via grant number EAR-1735676.

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
