# Peer review of "Cosmogenic nuclide dating of two stacked ice masses: Ong Valley, Antarctica"

_The Cryosphere, 2022_

## Author Response (AR1)

**Reviewer 1**

We appreciate the supportive and helpful comments by RC 1, and for most part agree with the recommendations. Please, see below our detailed responses (blue) to specific review comments.

**General comments**

This paper presents an interesting and novel data set from a mixed media (ice-sediment) core collected from an ice-free valley in the Transantarctic Mountains. The main aim of the study is to constrain the age of emplacement of the ice body/bodies found underneath a debris layer, an important topic as ancient glacial ice has high potential as a paleo-environmental archive. This study applies a brand new (to my knowledge) multi-nuclide approach to cosmogenically dating the ice body, or more specifically, the minerogenic debris on top and within it. The cosmogenic data-set is of high quality and importance and is without doubt worthy of publication.

The added novelty of this study comes from the application of forward model that predicts the evolution of nuclide concentrations with depth in a sublimating debris-rich ice mass given certain assumptions. This is a strong contribution to the cosmogenic toolbox and will undoubtedly be of interest to other researchers. The authors include well commented code that i was able to (mostly) follow and run myself. They should be commended for this.

My overall feeling is that this has potential to be a very influential paper, the finding that the ice is many millions of years old is exciting and will be of wide interest. I want to be supportive of this paper however i'm afraid to say i think the paper would benefit from a fairly significant re-write/re-structuring. The overall structure and clarity could be improved significantly and i think the different but complimentary approaches to constraining the age of the ice masses more clearly and logically articulated. Currently the descriptions of the overarching principles and their applications are scattered throughout the paper and i found it very difficult to follow.

**Specific comments**

I think there are several specific aspects of the paper and its structure that could be developed to improve the clarity. Firstly, I think the overall contextual information given regarding the core is not sufficient. The core clearly displays a varied stratigraphy, evident in both the visual appearance and measured debris content of the ice. This stratigraphy seems critical to the interpretation of the subsequent cosmogenic nuclide data but it is only very briefly described in the results section. The sub-division of the core into individual units should be based on the stratigraphy (i.e. descriptive) and not on the interpretation of the cosmo data (i.e. interpretative). It would also provide an opportunity to introduce some of the potential complexity within the core (e.g. potentially more than one ice mass/ice deposition event) given the observed stratigraphy. This would then seem to give a foundation for describing what units were sampled and why/what they might inform upon. I also think the core description should come much earlier in the paper, perhaps after the study site section, as is often done in ocean core studies.

We agree that the physical description of the ice core should be expanded and a natural location for that would be after the description of the field site as suggested. However, it does not change our general

approach to analyzing the core for three reasons: 1) originally, we started the analyses with the simplest assumption that the core covers a single glacial advance. The variations in the sediment concentration and the signs of banding that were observed in the core, are assumed to reflect the fact that the ice originates from near the bottom of the ice sheet and close to the perimeter of the glacier. 2) there are no major visible differences in the characteristics of the sediment in the core. 3) For our cosmogenic isotope analyses we used all mineral matter available in the core. In other words, our sampling scheme was not dictated or affected by the core physical characteristics.

The results section is very short and doesn't describe that data in enough detail for the reader to subsequently follow the paper. The cosmogenic data from the core is the central data-set but is described in less than five lines. The down core variation should be specifically described and quantified. The variations have implications for the subsequent application of the model.

We agree, and will expand and add more details to the results section.

Another comment regards the structure, both the overall structure and structure of individual sections. I appreciate that this may have been quite a tricky paper to write as it involves a number of approaches (surface exposure dating, burial dating, depth-profile dating) and modelling.

We agree wholeheartedly. This was a tricky paper to write.

I always find in similar cases structure can be hard to decide upon but one way I find helpful is to start with the simple(!) parts and add complexity. Currently it feels the paper sometimes tries to address all the complexity at once. For example in Section 4.3 the overarching principles of nuclide production at depth are described after the complexities of evolving mass shielding and depth. This seems like the wrong way round. Additionally, sentences describing key concepts are scattered throughout the paper; for example in section 6.2 (522-523, 526-536). The reader needs to be as up-to-speed as possible before the model results are introduced. I wonder whether the bulk of the model description should be moved to supplementary and only the key concepts described in the paper (perhaps in the discussion section).

The modeling approach and application are a novel contribution of this paper, therefore we feel that it needs to be in the body of the paper rather than in an appendix.

We agree that we need to provide the reader a clearer roadmap on how the modeling unfolds in the paper. In general, the organizing principle of this paper is that we want to clearly show how the unusual characteristics of the observations led us to choose methods of data analysis that we would not have initially expected to be useful. The use of burial dating to constrain the age of the ice is probably the best example. Describing burial dating in detail before the reader has seen the observations would be misleading to the reader, because it would imply that somehow we knew in advance of the study that burial dating would be a viable approach to determining the age of the ice. This is not the case – given only the geologic setting of the site and nothing else, neither we nor anyone else would expect that burial dating would be possible or useful. The usefulness of burial dating only became clear after the surprising observation that some of the englacial sediment must have experienced prior exposure before being entrained in the ice. The purpose of the paper organization was to make this chain of reasoning clear: there are many different ways to interpret cosmogenic-nuclide data to gain age information, the correct approach is not known in advance, and one has to choose the right approach based on both the geologic

context and the nature of the observations. However, it is true that we did not specifically state this organizing principle in the paper. We can improve this by stating this explicitly early in the paper.

In the paper the forward model is introduced in the order it was built. The debris concentrations and the density of the material are the foundations of the forward model. The shielding mass determines the production rate at depth. The production rate is introduced later since the production of cosmogenic nuclides are based on the change in depth/shielding mass that a sample experiences. We agree that this section would benefit from the addition of a paragraph in the beginning which states the basic principles of shielding and production rates of cosmogenic nuclides at depth, and the importance of shielding mass prior to the introduction of details involving shielding mass.

The burial dating is introduced after the forward model (section 4. Method) and the resulting nuclide data set (section 5. Results). The reasons for this are currently stated in section 3. (Cosmogenic-nuclide applications relevant for dating Ong Valley buried ice) and, we agree, should be described in greater detail. However, the burial dating is only necessary and applicable because of the downcore increase in nuclide concentrations, and is therefore introduced after the results which reveal this. We think that a paragraph describing the reasons to use the burial dating as a constraint would be beneficial in section 6, to help the reader to follow along.

Is burial dating part of the model? Line 526 suggests not but line 611 suggests it is? I am really confused. I think the model results and burial dating results need to be more clearly defined as to what is what. Section 6.2 has forward modelling in the title but seems to refer entirely to burial dating units that the forward model wasn't applied to? The burial ages are only given at the very end of section 6 even though they are used as constraints for the model?

The model is not itself a dating method, it is just a forward model calculation that predicts the nuclide concentrations we should observe as a function of various parameters including the age of the ice. The concept of burial dating comes into the model optimization because of the constraint that, in effect, the samples are not allowed to have a burial age less than zero at the time they are incorporated into the ice. Then in a second step, after we have identified a best-fitting model for the nuclide concentrations produced after ice emplacement, we compute apparent burial ages for samples from the recycled surface material units. Thus, the concept of burial dating is used in the model optimization, but the calculation of burial ages for some samples is a separate calculation that takes place in a subsequent step.

There are two benefits of using burial dating: 1) to constrain the model as any given sample within the ice core cannot have been buried for lesser time than the deposit of the ice that encloses it, hence this burial constraint will provide us with a maximum depositional age of the middle ice, and 2) to determine the burial age for all samples in order to evaluate whether there is a general agreement between the burial age of the samples or if there is variation in ages which would indicate a more complex history or a variable source of the debris. Therefore, the model results come first which include the burial dating constraint and is based on such, and later the remaining results of the burial dating that appear in section 6. However, we agree that the structure of such could be more clearly defined in the forward modeling section where burial dating is introduced and discussed.

I am clearly not following what was done from the text. I think a much clearer structure that separates the measured results from the modelled results is needed. They authors need to be explicit about what is what throughout the paper. The paper needs to set up a logical structure and follow it throughout, to me it currently jumps about from one approach to another making it really hard for me to follow and

subsequently review. The authors will of course be very familiar with the steps involved in deriving the results but for someone seeing this for the first time it is not obvious.

As noted above, the organizing principle of this paper is to show clearly how the characteristics of the data led us to choose approaches to data analysis. However, we agree that we have not specifically described this principle to the reader. We will explain this in more detail.

I would also ask that the authors think about some of the terminology used. The title describes dating of ice masses but the text commonly refers to ice mass (singular).  Similarly i dont think the term "middle" ice is helpful when talking about a vertical core with potentially multiple ice masses within it. To me the term paleo-surface implies it is in situ which i dont think the authors are implying; i think this links back to the point about sub-dividing the core on descriptive grounds not interpretative.

The "ice mass (singular/plural)" terminology is dictated by the fact that the description of the site starts with the simplest assumption of a single ice mass. The physical appearance of the core gave no clear indication of more than one ice mass, only after further analyses and modeling we learned that the core contained ice from two separate ice advances at this location. Therefore, we need to refer to them as ice masses.

Middle ice refers to the ice that is found directly below the middle drift that is exposed at the surface. The reason we need to use the qualifier "Middle" in the name is that there is also "Young ice" under the youngest drift located in the lower end of the valley. We need to explain this more clearly in section 2 (Study Area). We also think that capitalizing the names Middle Ice and Young Ice will help to identify them as proper nouns and not as adjectives.

In reference to the paleo surface, we do not imply that it is in situ. We will clarify this in the manuscript.

**Reviewer 2**

Thank you for your comments. We agree with all of Reviewer 2 comments and will make the appropriate changes and additions as suggested. Please find our specific replies (blue) in the below text.

**Summary**

This paper describes cosmogenic nuclide dating of buried ice masses in the Ong Valley, Antarctica, based on measurements from an almost 1000cm deep ice-sediment core. The main goal is to constrain the age of these (very old) ice masses. In order to achieve this, an exciting new forward model is presented that uses multiple cosmogenic nuclides measured in sediment samples throughout the core to constrain the age using some simple principles and assumptions. The dataset, model, and approach here will all obviously be important contributions to the scientific literature and I look forward to seeing these published. Although there are a few things that could be clarified (or reorganized?), in general this was a very rigorous treatment of the topic and all the information was provided here in order to fully understand the calculations and model (including available, well-commented code). The conclusions are based strongly on the data and the clearly stated assumptions and the conclusions are put fully in context of other publications and are aligned with the relevant uncertainty on the final results.

**Specific Comments**

At several points, I had questions on various things (density, grain sizes, steps in a process), but they all ended up getting covered later in the manuscript. However, the number of notes I had like this might mean that there is some reorganisation that could help.

L221-222: already introduced cosmo terminology (including proper superscripts), so probably easiest to follow that here.

This has now been fixed.

L277-278: The description of the factors that the supraglacial debris layer depends on clearly lists 4 factors, including concentration of debris in ice. The forward model presented immediately after this mirrors these factors except for the debris concentration. This was confusing until it gets explained significantly later in the paper. It would be useful to explain why this is not considered here (or that the 'missing parameter' will be explained later?).

Yes, we will address this in the text

Description of eqns (starting ~300): Not quite sure what all the subscripts were ($E_T$, $Z_T$) and the text wasn't consistent on parameters in italics, etc. (not huge, but slightly confusing). Could explain meaning up front (if relevant)

We agree that this was not consistent and this has now been fixed

L489: Describing two observations – 'the set of samples that display monotonically decreasing nuclide concentrations' was a bit confusing because I didn't initially realise that these were being considered

across the entire profile and not just within the different layers that had been identified (albeit identified using the nuclide concentrations?). Might help to specify.

We agree and will amend the text accordingly.

Figure 7: I absolutely love this figure! I did wish that I could see the zoom in of the S1 somewhere (to see the profile there). Also, the horizontal/vertical boxes (clearly visible in Ne the best) are not explained (I assumed horizontal width was uncertainty, but not sure what the different horizontal lines are? Divisions between samples that were combined?).

Thank you for pointing this out. We will add the description of the horizontal/vertical boxes to the figure caption. Yes, these do represent the measured uncertainty and division between samples that were combined.

L524: Not sure what the surface sample is referring to – S1, E1 and E2 are all perfectly clear here. Perhaps refer back to the appropriate section/table since this hasn't been recently discussed.

The surface sample is included in S1 and should not be listed individually. This has been fixed now

L546: Are there samples that are not used for forward model fitting that are NOT in the recycled surface material? Not sure if I missed something here...

No samples are excluded. Any samples not used for model fitting are used as burial constraints. We will add a statement to clarify this.

Figure 11: The red dots are very hard to see (tiny and almost covered by label text).

Thank you for pointing this out, it has been fixed.

The majority of the paper is very easy to understand, but there are some sections where a bit of editing might help readability (extra commas needed, small edits to grammar: e.g. L529 'as follows' instead of is following). Nothing huge, but a few times where I had to read a sentence twice to figure out clauses, etc.